# ON THE IMPOSSIBILITY OF GLOBAL CONVERGENCE IN MULTI-LOSS OPTIMIZATION

**Alistair Letcher**
aletcher.github.io

## ABSTRACT

Under mild regularity conditions, gradient-based methods converge globally to a critical point in the single-loss setting. This is known to break down for vanilla gradient descent when moving to *multi-loss* optimization, but can we hope to build some algorithm with global guarantees? We negatively resolve this open problem by proving that desirable convergence properties cannot simultaneously hold for *any* algorithm. Our result has more to do with the existence of games with no satisfactory outcomes, than with algorithms per se. More explicitly we construct a two-player game with zero-sum interactions whose losses are both coercive and analytic, but whose only simultaneous critical point is a strict maximum. Any 'reasonable' algorithm, defined to avoid strict maxima, will therefore fail to converge. This is fundamentally different from single losses, where coercivity implies existence of a global minimum. Moreover, we prove that a wide range of existing gradient-based methods almost surely have bounded but non-convergent iterates in a constructed *zero-sum* game for suitably small learning rates. It nonetheless remains an open question whether such behavior can arise in high-dimensional games of interest to ML practitioners, such as GANs or multi-agent RL.

## 1 INTRODUCTION

**Problem Setting.**   As multi-agent architectures proliferate in machine learning, it is becoming increasingly important to understand the dynamics of gradient-based methods when optimizing multiple interacting goals, otherwise known as *differentiable games*. This framework encompasses GANs (Goodfellow et al., 2014), intrinsic curiosity (Pathak et al., 2017), imaginative agents (Racanière et al., 2017), synthetic gradients (Jaderberg et al., 2017), hierarchical reinforcement learning (Wayne & Abbott, 2014; Vezhnevets et al., 2017) and multi-agent RL in general (Busoniu et al., 2008). The interactions between learning agents make for vastly more complex mechanics: naively applying gradient descent on each loss simultaneously is known to diverge even in simple bilinear games.

**Related Work.**   A large number of methods have recently been proposed to alleviate the failings of simultaneous gradient descent: adaptations of single-loss algorithms such as Extragradient (EG) (Azizian et al., 2019) and Optimistic Mirror Descent (OMD) (Daskalakis et al., 2018), Alternating Gradient Descent (AGD) for finite regret (Bailey et al., 2019), Consensus Optimization (CO) for GAN training (Mescheder et al., 2017), Competitive Gradient Descent (CGD) based on solving a bilinear approximation of the loss functions (Schaefer & Anandkumar, 2019), Symplectic Gradient Adjustment (SGA) based on a novel decomposition of game mechanics (Balduzzi et al., 2018; Letcher et al., 2019a), and opponent-shaping algorithms including Learning with Opponent-Learning Awareness (LOLA) (Foerster et al., 2018) and its convergent counterpart, Stable Opponent Shaping (SOS) (Letcher et al., 2019b). Let $\mathcal{A}$ be this set of algorithms.

Each has shown promising theoretical implications and empirical results, but none offers insight into global convergence in the *non-convex* setting, which includes the vast majority of machine learning applications. One of the main roadblocks compared with single-loss optimization has been noted by Schaefer & Anandkumar (2019): "a convergence proof in the nonconvex case analogue to Lee et al. (2016) is still out of reach in the competitive setting. A major obstacle to this end is the identification of a suitable measure of progress (which is given by the function value in the single agent setting), since norms of gradients can not be expected to decay monotonously for competitive dynamics in non-convex-concave games."

It has been established that Hamiltonian Gradient Descent converges in two-player zero-sum games under a "sufficiently bilinear" condition by Abernethy et al. (2019), but this algorithm is unsuitable for optimization as it cannot distinguish between minimization and maximization (Hsieh et al., 2020, Appendix C.4). Global convergence has also been established for some algorithms in a few special cases: potential and Hamiltonian games (Balduzzi et al., 2018), zero-sum games satisfying the two-sided Polyak-Łojasiewicz condition (Yang et al., 2020), zero-sum linear quadratic games (Zhang et al., 2019) and zero-sum games whose loss and first three derivatives are bounded (Mangoubi & Vishnoi, 2020). These are significant contributions with several applications of interest, but do not include any of the architectures mentioned above. Finally, Balduzzi et al. (2020) show that GD dynamics are bounded under a 'negative sentiment' assumption in smooth markets, which do include GANs – but this does not imply convergence, as we will show.

On the other hand, *failure* of global convergence has been shown for the Multiplicative Weights Update method by Palaiopanos et al. (2017), for policy-gradient algorithms by Mazumdar et al. (2020), and for simultaneous and alternating gradient descent (simGD and AGD) by Vlatakis-Gkaragkounis et al. (2019); Bailey et al. (2019), with interesting connections to Poincaré recurrence. Nonetheless, nothing is claimed about other optimization methods. Farnia & Ozdaglar (2020) show that GANs may have no Nash equilibria, but it does *not* follow that algorithms fail to converge since there may be locally-attracting but non-Nash critical points (Mazumdar et al., 2019, Example 2).

Finally, Hsieh et al. (2020) uploaded a preprint just after the completion of this work with a similar focus to ours. They prove that generalized Robbins-Monro schemes may converge with arbitrarily high probability to spurious attractors. This includes simGD, AGD, stochastic EG, optimistic gradient and Kiefer-Wolfowitz. However, Hsieh et al. (2020) focus on the possible occurrence of undesirable convergence phenomena for *stochastic* algorithms. We instead prove that desirable convergence properties cannot simultaneously hold for all algorithms (including *deterministic*). Moreover, their results apply only to decreasing step-sizes whereas ours include constant step-sizes. These distinctions are further highlighted by Hsieh et al. (2020) in the further related work section. Taken together, our works give a fuller picture of the failure of global convergence in multi-loss optimization.

**Contribution.** We prove that global convergence in multi-loss optimization is fundamentally incompatible with the 'reasonable' requirement that algorithms avoid strict maxima and converge only to critical points. We construct a two-player game with zero-sum interactions whose losses are coercive and analytic, but whose only critical point is a strict maximum (Theorem 1). Reasonable algorithms must either diverge to infinite losses or cycle (bounded non-convergent iterates).

One might hope that global convergence could at least be guaranteed in games with strict *minima* and no other critical points. On the contrary we show that strict minima can have arbitrarily small regions of attraction, in the sense that reasonable algorithms will fail to converge there with arbitrarily high probability for fixed initial parameter distribution (Theorem 2).

Finally, restricting the game class even further, we construct a *zero-sum* game in which all algorithms in $\mathcal{A}$ (as defined in Appendix A) are proven to cycle (Theorem 3).

It may be that cycles do not arise in high-dimensional games of interest including GANs. Proving or disproving this is an important avenue for further research, but requires that we recognise the impossibility of global guarantees in the first place.

## 2 BACKGROUND

### 2.1 SINGLE LOSSES: GLOBAL CONVERGENCE OF GRADIENT DESCENT

Given a continuously differentiable function $f : \mathbb{R}^d \to \mathbb{R}$, let

$$\theta_{k+1} = \theta_k - \alpha \nabla f(\theta_k)$$

be the iterates of gradient descent with learning rate $\alpha$, initialised at $\theta_0$. Under standard regularity conditions, gradient descent converges globally to critical points:

**Proposition 1.** *Assume $f \in C^2$ has compact sublevel sets and is either analytic or has isolated critical points. For any $\theta_0 \in \mathbb{R}^d$, define $U_0 = \{f(\theta) \leq f(\theta_0)\}$ and let $L < \infty$ be a Lipschitz constant for $\nabla f$ in $U_0$. Then for any $0 < \alpha < 2/L$ we have $\lim_k \theta_k = \bar{\theta}$ for some critical point $\bar{\theta}$.*

The requirements for convergence are relatively mild:

1. $f$ has compact sublevel sets iff $f$ is coercive, $\lim_{\|\theta\| \to \infty} f(\theta) = \infty$, which mostly holds in machine learning since $f$ is a loss function.

2. $f$ has isolated critical points if it is a Morse function (nondegenerate Hessian at critical points), which holds for almost all $C^2$ functions. More precisely, Morse functions form an open, dense subset of all functions $f \in C^2(\mathbb{R}^d, \mathbb{R})$ in the Whitney $C^2$-topology.

3. Global Lipschitz continuity is *not* assumed, which would fail even for cubic polynomials.

The goal of this paper is to prove that similar (even weaker) guarantees *cannot* be obtained in the multi-loss setting – not only for GD, but for any reasonable algorithm. This has to do with the more complex nature of gradient vector fields arising from multiple losses.

## 2.2 DIFFERENTIABLE GAMES

Following Balduzzi et al. (2018), we frame the problem of multi-loss optimization as a differentiable game among cooperating and competing agents/players. These may simply be different internal components of a single system, like the generator and discriminator in GANs.

**Definition 1.** *A differentiable game is a set of $n$ agents with parameters $\theta = (\theta^1, \ldots, \theta^n) \in \mathbb{R}^d$ and twice continuously differentiable losses $L^i : \mathbb{R}^d \to \mathbb{R}$, where $\theta^i \in \mathbb{R}^{d_i}$ for each $i$ and $\sum_i d_i = d$.*

Losses are *not* assumed to be convex/concave in any of the parameters. In practice, losses need only be differentiable almost-everywhere: think of neural nets with rectified linear units.

If $n = 1$, the 'game' is simply to minimise a given loss function. We write $\nabla_i L^k = \nabla_{\theta^i} L^k$ and $\nabla_{ij} L^k = \nabla_{\theta^j} \nabla_{\theta^i} L^k$ for any $i, j, k$, and define the simultaneous gradient of the game

$$\xi = \left(\nabla_1 L^1, \ldots, \nabla_n L^n\right)^T \in \mathbb{R}^d$$

as the concatenation of each player's gradient. If each agent independently minimises their loss using GD with learning rate $\alpha$, the parameter update for all agents is given by $\theta \leftarrow \theta - \alpha\xi(\theta)$. We call this simultaneous gradient descent (simGD), or GD for short. We call $\bar{\theta}$ a *critical point* if $\xi(\bar{\theta}) = 0$. Now introduce the 'Hessian' (or Jacobian) of the game as the block matrix

$$H = \nabla\xi = \begin{pmatrix} \nabla_{11} L^1 & \cdots & \nabla_{1n} L^1 \\ \vdots & \ddots & \vdots \\ \nabla_{n1} L^n & \cdots & \nabla_{nn} L^n \end{pmatrix} \in \mathbb{R}^{d \times d}.$$

Importantly note that $H$ is not symmetric in general unless $n = 1$, in which case we recover the usual Hessian $H = \nabla^2 L$. However $H$ can be decomposed into symmetric and anti-symmetric components as $H = S + A$ (Balduzzi et al., 2018). A second useful decomposition has appeared recently in (Letcher et al., 2019b) and (Schaefer & Anandkumar, 2019): $H = H_d + H_o$ where $H_d$ and $H_o$ are the matrices of diagonal and off-diagonal blocks; formally, $H_d = \bigoplus_i \nabla_{ii} L^i$. One solution concept for differentiable games, analogous to the single-loss case, is defined as follows.

**Definition 2.** *A critical point $\bar{\theta}$ is a (strict, local) minimum if $H(\bar{\theta}) \succ 0$.*[1]

These were named (strict) stable fixed points by Balduzzi et al. (2018), but the term is usually reserved in dynamical systems to the larger class defined by Hessian eigenvalues with positive real parts, which is implied but not equivalent to $H \succ 0$ for non-symmetric matrices.

In particular, strict minima are (differential) Nash equilibria as defined by Mazumdar et al. (2019), since diagonal blocks must also be positive definite: $\nabla_{ii} L^i(\bar{\theta}) \succ 0$. The converse does not hold.

**Algorithm class.** This paper is concerned with any algorithm whose iterates are obtained by initialising $\theta_0$ and applying a function $F$ to the previous iterates, namely $\theta_{k+1} = F(\theta_k, \ldots, \theta_0)$. This holds for all gradient-based methods (deterministic or stochastic); most of them are only functions

---

[1] For non-symmetric matrices, positive definiteness is defined as $H \succ 0$ iff $u^T H u > 0$ for all non-zero $u \in \mathbb{R}^d$. This is equivalent to the symmetric part $S$ of $H$ being positive definite.

of the current iterate $\theta_k$, so that $\theta_k = F^k(\theta_0)$. All probabilistic statements in this paper assume that $\theta_0$ is initialised following any bounded and continuous measure $\nu$ on $\mathbb{R}^d$. Continuity is a weak requirement and widely holds across machine learning, while boundedness mostly holds in practice since the bounded region can be made large enough to accommodate required initial points.

For single-player games, the goal of such algorithms is for $\theta_k$ to converge to a local (perhaps global) minimum as $k \to \infty$. The goal is less clear for differentiable games, but is generally to reach a minimum or a Nash equilibrium. In the case of GANs the goal might be to reach parameters that produce realistic images, which is more challenging to define formally.

Throughout the text we use the term *(limit) cycle* to mean *bounded but non-convergent iterates*. This terminology is used because bounded iterates are non-convergent if and only if they have at least two accumulation points, between which they must 'cycle' infinitely often. This is not to be taken literally: the set of accumulation points may not even be connected. Hsieh et al. (2020) provide a more complete characterisation of these cycles.

**Game class.**  Expecting global guarantees in *all* differentiable games is excessive, since every continuous dynamical system arises as simultaneous GD on the loss functions of a differentiable game (Balduzzi et al., 2020, Lemma 1). For this reason, the aforementioned authors have introduced a vastly more tractable class of games called *markets*.

**Definition 3.** *A (smooth) market is a differentiable game where interactions between players are pairwise zero-sum, namely,*

$$L^i(\theta) = L^i(\theta^i) + \sum_{j \neq i} g_{ij}(\theta^i, \theta^j)$$

*with $g_{ij}(\theta^i, \theta^j) + g_{ji}(\theta^j, \theta^i) = 0$ for all $i, j$.*

This generalises zero-sum games while remaining amenable to optimization and aggregation, meaning that "we can draw conclusions about the gradient-based dynamics of the collective by summing over properties of its members" (Balduzzi et al., 2020). Moreover, this class captures a large number of applications including GANs and related architectures, intrinsic curiosity modules, adversarial training, task-suites and population self-play. One would modestly hope for *some* reasonable algorithm to converge globally in markets. We will prove that even this is too much to ask.

### 2.3  REASONABLE ALGORITHMS

We wish to prove that global convergence is at odds with weak, 'reasonable' desiderata. The first requirement is that fixed points of an optimization algorithm $F$ are critical points. Formally,

$$F(\theta) = \theta \implies \xi(\theta) = 0. \tag{R1}$$

If not, some agent $i$ could strictly improve its losses by following the gradient $-\nabla_i L^i \neq 0$. There is no reason for a gradient-based algorithm to stop improving if its gradient is non-zero.

The second requirement is that algorithms avoid strict maxima. Analogous to strict minima, they are defined for single losses by a negative-definite Hessian $H \prec 0$. Converging to such a point $\bar{\theta}$ is the opposite goal of any meaningful algorithm since moving *anywhere* away from $\bar{\theta}$ decreases the loss. There are multiple ways of generalising this concept for multiple losses, but Proposition 2 below justifies that $H \prec 0$ is the weakest one.

**Proposition 2.** *Write $\lambda(A) = \mathrm{Re}(\mathrm{Spec}(A))$ for real parts of the eigenvalues of a matrix $A$. We have the following implications, and none of them are equivalences.*

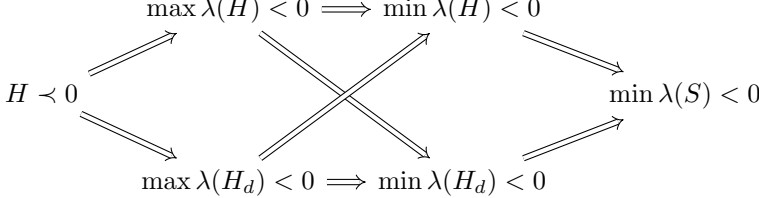

**Definition 4.** *A critical point $\bar{\theta}$ is a (strict, local)* maximum *if $H(\bar{\theta}) \prec 0$.*

Imposing that algorithms avoid strict maxima is therefore the weakest possible requirement of its kind. Note that the bottom-left implication Proposition 2 is equivalent to $\nabla_{ii}L^i \prec 0$ for all $i$, so strict maxima are also strict maxima of each player's individual loss function. Players can *all* decrease their losses by moving *anywhere* away from them. It is exceedingly reasonable to ask that optimization algorithms avoid these points almost surely. Formally, we require that for any strict maximum $\bar{\theta}$ and bounded region $U$ there are hyperparameters such that

$$\mu\left(\{\theta_0 \in U \mid \lim_k \theta_k = \bar{\theta}\}\right) = 0. \tag{R2}$$

$\mu$ denotes Lebesgue measure. Hyperparameters may depend on the given game and the region $U$, as is typical for learning rates in gradient-based methods.

**Definition 5** (Reason). *An algorithm is* reasonable *if it satisfies R1 and R2.*

Reason is not equivalent to *rationality* or *self-interest*. Reason is much weaker, imposing only that agents are well-behaved regarding strict maxima even if their individual behavior is not self-interested. For instance, SGA agents do not behave out of self-interest (Balduzzi et al., 2018).

## 3 GLOBAL CONVERGENCE IN DIFFERENTIABLE GAMES

### 3.1 REASONABLE ALGORITHMS FAIL TO CONVERGE GLOBALLY

Our main contribution is to show that global guarantees do not exist for *any* reasonable algorithm. First recall that global convergence should not be expected in *all* games, since there may be a divergent direction with minimal loss (imagine minimising $L = e^x$). It should however be asked that algorithms have bounded iterates in *coercive* games, defined by coercive losses

$$\lim_{\|\theta\| \to \infty} L^i(\theta) = \infty$$

for all $i$. Indeed, unbounded iterates in coercive games would lead to infinite losses for *all* agents, the worst possible outcome. Given bounded iterates, convergence should hold if the Hessian is nondegenerate at critical points (which must therefore be isolated, recall Proposition 1). We call such a game *nondegenerate*. This condition can also be replaced by analyticity of the loss. In the spirit of weakest assumptions, we ask for convergence when *both* conditions hold.

**Definition 6** (Globality). *An algorithm is* global *if, in a coercive, analytic and nondegenerate game, for any fixed $\theta_0$, iterates $\theta_k$ are bounded and converge for suitable hyperparameters.* (G1)

Note that GD is global for *single-player* games by Proposition 1. Unfortunately, reason and globality are fundamentally at odds as soon as we move to two-player markets.

**Theorem 1.** *There is a coercive, nondegenerate, analytic two-player market $\mathcal{M}$ whose only critical point is a strict maximum. In particular, algorithms only have four possible outcomes in $\mathcal{M}$:*

1. *Iterates are unbounded, and all players diverge to infinite loss. [Not global]*

2. *Iterates are bounded and converge to the strict maximum. [Not reasonable]*

3. *Iterates are bounded and converge to a non-critical point. [Not reasonable]*

4. *Iterates are bounded but do not converge (cycle). [Not global]*

*Proof.* Consider the analytic market $\mathcal{M}$ given by

$$L^1(x,y) = x^6/6 - x^2/2 + xy + \frac{1}{4}\left(\frac{y^4}{1+x^2} - \frac{x^4}{1+y^2}\right)$$

$$L^2(x,y) = y^6/6 - y^2/2 - xy - \frac{1}{4}\left(\frac{y^4}{1+x^2} - \frac{x^4}{1+y^2}\right).$$

We prove in Appendix D that $\mathcal{M}$ is coercive, nondegenerate, and has a unique critical point at the origin, which is a strict maximum. $\qquad\square$

Constructing an algorithm with global guarantees is therefore doomed to be unreasonable in that it will converge to strict maxima or non-critical points in $\mathcal{M}$.

None of the outcomes of $\mathcal{M}$ are satisfactory. The first three are highly objectionable, as already discussed. The fourth is less obvious, and may even have game-theoretic significance (Papadimitriou & Piliouras, 2019), but is counter-intuitive from an optimization standpoint. Terminating the iteration would lead to a non-critical point, much like the third outcome. Even if we let agents update parameters continuously as they play a game or solve a task, they will have oscillatory behavior and fail to produce consistent outcomes (e.g. when generating an image or playing Starcraft).

The hope for machine learning is that such predicaments do not arise in applications we care about, such as GANs or intrinsic curiosity. This may well be the case, but proving or disproving global convergence in these specific settings is beyond the scope of this paper.

**Remark.** Why can this approach not be used to *disprove* global convergence for single losses? One reason is that we cannot construct a coercive loss with no critical points other than strict maxima: coercive losses, unlike games, always have a global minimum.

### 3.2    WHAT IF THERE ARE STRICT MINIMA?

One might wonder if it is purely the absence of strict minima that causes non-convergence, since strict minima are locally attracting under gradient dynamics. Can we guarantee global convergence if we impose existence of a minimum, and more, the absence of any other critical points?

Unfortunately, strict minima may have an arbitrarily small region of attraction. Assuming parameters are initialised following any bounded continuous measure $\nu$ on $\mathbb{R}^d$, we can always modify $\mathcal{M}$ by deforming a correspondingly small region around the origin, turning it into a minimum while leaving the dynamics unchanged outside of this region.

For a fixed initial distribution, any reasonable algorithm can therefore enter a limit cycle or diverge to infinite losses with arbitrarily high probability.

**Theorem 2.** *Given a reasonable algorithm with bounded continuous distribution on $\theta_0$ and a real number $\epsilon > 0$, there exists a coercive, nondegenerate, almost-everywhere analytic two-player market $\mathcal{M}_\sigma$ with a strict minimum and no other critical points, such that $\theta_k$ either cycles or diverges to infinite losses for both players with probability at least $1 - \epsilon$.*

*Proof.* Let $0 < \sigma < 0.1$ and define

$$f_\sigma(\theta) = \begin{cases} (x^2 + y^2 - \sigma^2)/2 & \text{if } \|\theta\| \geq \sigma \\ (y^2 - 3x^2)(x^2 + y^2 - \sigma^2)/(2\sigma^2) & \text{otherwise,} \end{cases}$$

where $\theta = (x, y)$ and $\|\theta\| = \sqrt{x^2 + y^2}$ is the standard $L2$-norm. Note that $f_\sigma$ is continuous since

$$\lim_{\|\theta\| \to \sigma^+} f_\sigma(x, y) = 0 = \lim_{\|\theta\| \to \sigma^-} f_\sigma(x).$$

Now consider the two-player market $\mathcal{M}_\sigma$ given by

$$L^1(x, y) = x^6/6 - x^2 + f_\sigma(x, y) + xy + \frac{1}{4}\left(\frac{y^4}{1 + x^2} - \frac{x^4}{1 + y^2}\right)$$

$$L^2(x, y) = y^6/6 - f_\sigma(x, y) - xy - \frac{1}{4}\left(\frac{y^4}{1 + x^2} - \frac{x^4}{1 + y^2}\right).$$

We prove in Appendix E that $\mathcal{M}_\sigma$ is a coercive, nondegenerate, almost-everywhere analytic game whose only critical point is a strict minimum at the origin. We then prove that $\theta_k$ cycles or diverges with probability at least $1 - \epsilon$, and plot iterates for each algorithm in $\mathcal{A}$. □

### 3.3    HOW DO EXISTING ALGORITHMS BEHAVE?

Any algorithm will either fail to be reasonable or global in $\mathcal{M}$. Nonetheless, it would be interesting to determine the specific failure that each algorithm in $\mathcal{A}$ exhibits. Each of them is defined in

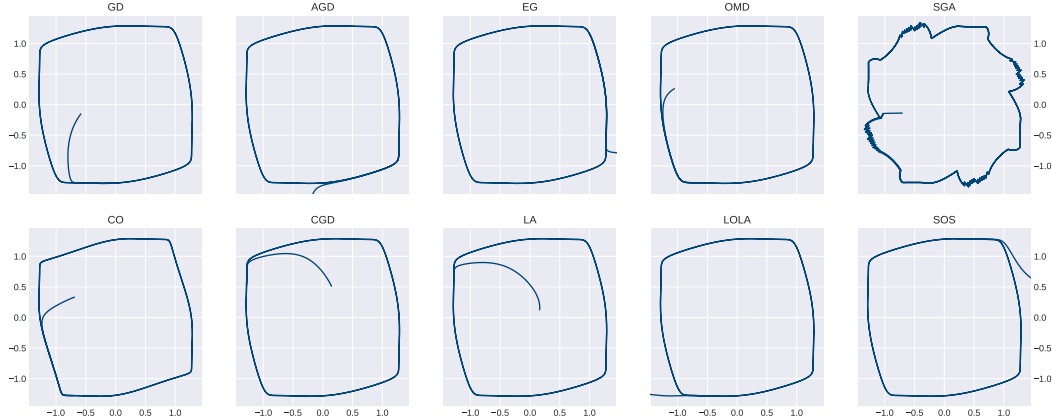

Figure 1: Algorithms in $\mathcal{A}$ fail to converge in $\mathcal{M}$ with $\alpha = \gamma = 0.01$. Single run with standard normal initialisation, 3000 iterations. The behavior of SGA is slightly different, explained by the presence of a non-continuous parameter $\lambda$ jumping between $\pm 1$ according to an alignment criterion.

Appendix A, writing $\alpha$ for the learning rate and $\gamma$ for the Consensus Optimization hyperparameter. We expect each algorithm to be reasonable and moreover to have bounded iterates in $\mathcal{M}$ for suitably small hyperparameters. If this holds, they must cycle by Theorem 1.

This was witnessed experimentally across 1000 runs for $\alpha = \gamma = 0.01$, with every run resulting in cycles. A single such run is illustrated in Figure 1. Algorithms may follow one of the three other outcomes for other hyperparameters, for instance diverging to infinite loss if $\alpha$ is too large or converging to the strict maximum for CO if $\gamma$ is too large. The point here is to characterise the 'regular' behavior which can be seen as that occurring for sufficiently small hyperparameters.

Instead of *proving* that algorithms must cycle in $\mathcal{M}$, we construct a *zero-sum* game $\mathcal{N}$ with similar properties as $\mathcal{M}$ and prove below that algorithms in $\mathcal{A}$ almost surely fail to converge there for small $\alpha, \gamma$. This is stronger than proving the analogous result for $\mathcal{M}$, since $\mathcal{N}$ belongs to the even smaller class of zero-sum games which one might have hoped was well-behaved.

In this light, one might wish to extend Theorem 1 to zero-sum games. However, zero-sum games cannot be coercive since $L^1 \to \infty$ implies $L^2 \to -\infty$. It is therefore unclear whether global guarantees should be expected. Note however that $\mathcal{N}$ will be *weakly-coercive* in the sense that

$$\lim_{\|\theta^i\| \to \infty} L^i(\theta^i, \theta^{-i}) = \infty$$

for all $i$ and fixed $\theta^{-i}$.

**Theorem 3.** *There is a weakly-coercive, nondegenerate, analytic two-player zero-sum game $\mathcal{N}$ whose only critical point is a strict maximum. Algorithms in $\mathcal{A}$ almost surely have bounded non-convergent iterates in $\mathcal{N}$ for $\alpha, \gamma$ sufficiently small.*

*Proof.* Consider the analytic zero-sum game $\mathcal{N}$ given by

$$L^1 = xy - x^2/2 + y^2/2 + x^4/4 - y^4/4 = -L^2 .$$

We prove in Appendix F that $\mathcal{N}$ is weakly-coercive, nondegenerate, and has a unique critical point at the origin which is a strict maximum. We prove that algorithms in $\mathcal{A}$ have the origin as unique fixed points, with negative-definite Jacobian for $\alpha, \gamma$ small, hence failing to converge almost surely. We moreover prove that algorithms have *bounded* non-convergent iterates in $\mathcal{N}$ for $\alpha, \gamma$ sufficiently small. Iterates are plotted for a single run of each algorithm in Figure 3 with $\alpha = \gamma = 0.01$. $\square$

As in $\mathcal{M}$, the behavior of each algorithm may differ for larger hyperparameters. All algorithms may have unbounded iterates or converge to the strict maximum for large $\alpha$, while EG and OMD may even converge to a non-critical point (see proof). All such outcomes are unsatisfactory, though unbounded iteration will not result in positive infinite losses for *both* players since $L^1 = -L^2$.

### 3.4 COROLLARY: THERE ARE NO SUITABLE MEASURES OF PROGRESS

A crucial step in proving global convergence of GD on single losses is showing that the set of accumulation points is a subset of critical points, using the function value as a 'measure of progress'. The fact that this fails for differentiable games implies that there can be no suitable measures of progress for reasonable algorithms with bounded iterates. We formalise this below, answering the question of Schaefer & Anandkumar (2019) quoted in the introduction.

**Definition 7.** *A measure of progress for an algorithm given by $\theta_{k+1} = F(\theta_k)$ is a continuous map $M : \mathbb{R}^d \to \mathbb{R}$, bounded below, such that $M(F(\theta)) \leq M(\theta)$ and $M(F(\theta)) = M(\theta)$ iff $F(\theta) = \theta$.*

Measures of progress are very similar to *descent functions*, as defined by Luenberger & Ye (1984), and somewhat akin to Lyapunov functions. The function value $f$ is a measure of progress for single-loss GD under the usual regularity conditions, while the gradient norm $\|\xi\|$ is a measure of progress for GD in *strictly convex* differentiable games:

$$\|\xi(\theta - \alpha\xi)\|^2 = \|\xi\|^2 - \alpha\xi^T H^t \xi + o(\alpha) \leq \|\xi\|^2$$

for small $\alpha$. Unfortunately, games like $\mathcal{M}$ prevent the existence of such measures in general.

**Corollary 1.** *There are no measures of progress for reasonable algorithms which produce bounded iterates in $\mathcal{M}$ or $\mathcal{N}$.*

Assuming the algorithm to be reasonable is necessary: any map is a measure of progress for the unreasonable algorithm $F(\theta) = \theta$. Assuming the algorithm to have bounded iterates in $\mathcal{M}$ or $\mathcal{N}$ is necessary: $M(\theta) = \exp(-\theta \cdot \mathbf{1})$ is a measure of progress for the reasonable but always-divergent algorithm $F(\theta) = \theta + \mathbf{1}$, where $\mathbf{1}$ is the constant vector of ones.

## 4 CONCLUSION

We have proven that global convergence is fundamentally at odds with weak, desirable requirements in multi-loss optimization. Any reasonable algorithm can cycle or diverge to infinite losses, even in two-player markets. This arises because coercive games, unlike losses, may have no critical points other than strict maxima. However, this is not the only point of failure: strict minima may have arbitrarily small regions of attraction, making convergence arbitrarily unlikely.

Limit cycles are not necessarily bad: they may even have game-theoretic significance (Papadimitriou & Piliouras, 2019). This paper nonetheless shows that some games have no satisfactory outcome in the usual sense, even in the class of two-player markets. Players should neither escape to infinite losses, nor converge to strict maxima or non-critical points, so cycling may be the lesser evil. The community is accustomed to optimization problems whose solutions are single points, but cycles may have to be accepted as solutions in themselves.

The hope for machine learning practitioners is that local minima with large regions of attraction prevent limit cycles from arising in applications of interest, including GANs. Proving or disproving this is an interesting and important avenue for further research, with real implications on what to expect when agents learn while interacting with others. Cycles may for instance be unacceptable in self-driving cars, where oscillatory predictions may have life-threatening implications.

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

# APPENDIX

## A  ALGORITHMS AND EXPERIMENT HYPERPARAMETERS

Each algorithm in $\mathcal{A}$ cited in the 'Related Work' section can be defined as $F(\theta) = \theta - \alpha G(\theta)$ for some continuous $G : \mathbb{R}^d \to \mathbb{R}^d$. We have already seen that simultaneous GD is given by $G_{GD} = \xi$. The only examples in this paper are two-player games, for which AGD is given by

$$G_{\text{AGD}} = \begin{pmatrix} \xi_1(\theta_1, \theta_2) \\ \xi_2(\theta_1 - \alpha\xi_1, \theta_2) \end{pmatrix}$$

The other algorithms are given by

$$G_{\text{EG}} = \xi \circ (\text{id} - \alpha\xi) \qquad\qquad G_{\text{OMD}} = 2\xi(\theta_k) - \xi(\theta_{k-1})$$

$$G_{\text{SGA}} = (I + \lambda A^T)\xi \qquad\qquad G_{\text{CO}} = (I + \gamma H^T)\xi$$

$$G_{\text{CGD}} = (I + \alpha H_o)^{-1}\xi \qquad\qquad G_{\text{LA}} = (I - \alpha H_o)\xi$$

$$G_{\text{LOLA}} = (I - \alpha H_o)\xi - \alpha \operatorname{diag}(H_o^T \nabla L) \qquad G_{\text{SOS}} = (I - \alpha H_o)\xi - p\alpha \operatorname{diag}(H_o^T \nabla L) \,.$$

For OMD, the previous iterate can be uniquely recovered as $\theta_{k-1} = (\text{id} - \alpha\xi)^{-1}(\theta_k)$ using the proximal point algorithm if $\|H\| \le L$ and $\alpha < 1/L$, giving

$$G_{OMD} = 2\xi - \xi \circ (\text{id} - \alpha\xi)^{-1} \,.$$

In all experiments we initialise $\theta_0$ following a standard normal distribution and use a learning rate $\alpha = 0.01$, with $\gamma = 0.01$ for CO. Learning rates $\alpha_i$ could be chosen to be different for each player $i$, but we set them to be equal throughout this paper for simplicity. Claims regarding the behavior of each algorithm for sufficiently small $\alpha$ mean that all $\alpha_i$ should be sufficiently small. The $\lambda$ parameter for SGA is obtained by the alignment criterion introduced in the original paper,

$$\lambda = \text{sign}\left(\langle \xi, H^T\xi\rangle\langle A^T\xi, H^T\xi\rangle\right) \,.$$

Similarly, the $p$ parameter for SOS is given by a two-part criterion which need not be described here.

Accompanying code for all experiments can be found at https://github.com/aletcher/impossibility-global-convergence.

## B  PROOF OF PROPOSITION 1

We first prove a lemma and state a standard optimization result.

**Lemma 0.** *Let $G \in C^1(U, \mathbb{R}^d)$ for an open set $U$. If $G$ is $L$-Lipschitz then $\sup_{\theta \in U} \|\nabla G(\theta)\| \le L$.*

The proof is an adaptation of (Panageas & Piliouras, 2017, Lemma 7) for non-convex sets.

*Proof.* Fix any $\theta \in U$ and $\epsilon > 0$. Since $U$ is open, the ball $B_r(\theta)$ of radius $r$ centered at $\theta$ is contained in $U$ for some $r > 0$. By Taylor expansion, for any unit vector $\theta'$,

$$\|G(\theta + r\theta') - G(\theta)\| \ge r\|\nabla G(\theta)\theta'\| - o(r) \ge r\|\nabla G(\theta)\theta'\| - \epsilon r$$

for $r$ sufficiently small. Since $G$ is $L$-Lipschitz, we obtain

$$r\|\nabla G(\theta)\theta'\| \le \|G(\theta + r\theta') - G(\theta)\| + r\epsilon \le r(L + \epsilon) \,.$$

Since $\epsilon$ was arbitrary, $\|\nabla G(\theta)\theta'\| \le L$ for any unit $\theta'$. By definition of the norm, we obtain

$$\|\nabla G(\theta)\| = \sup_{\|\theta'\|=1} \|\nabla G(\theta)\theta'\| \le L$$

for all $\theta \in U$ and hence $\sup_{\theta \in U} \|\nabla G(\theta)\| \le L$. $\qquad\qquad\square$

**Proposition** ((Lange, 2013, Prop. 12.4.4) and (Absil et al., 2005, Th. 4.1)). *Assume $f$ has $L$-Lipschitz gradient and is either analytic or has isolated critical points. Then for any $0 < \alpha < 2/L$ and $\theta_0 \in \mathbb{R}^d$ we have*

$$\lim_k \|\theta_k\| = \infty \quad or \quad \lim_k \theta_k = \bar{\theta}$$

*for some critical point $\bar{\theta}$. If $f$ moreover has compact sublevel sets then the latter holds, $\lim_k \theta_k = \bar{\theta}$.*

We can now prove Proposition 1, which avoids requiring Lipschitz continuity by proving that iterates are contained in the sublevel set given by $\theta_0$ for appropriate learning rate $\alpha$.

**Proposition 1.** *Assume $f \in C^2$ has compact sublevel sets and is either analytic or has isolated critical points. For any $\theta_0 \in \mathbb{R}^d$, define $U_0 = \{f(\theta) \leq f(\theta_0)\}$ and let $L < \infty$ be a Lipschitz constant for $\nabla f$ in $U_0$. Then for any $0 < \alpha < 2/L$ we have $\lim_k \theta_k = \bar{\theta}$ for some critical point $\bar{\theta}$.*

*Proof.* Note that $\nabla f \in C^1$, so $f$ has $L$-Lipschitz gradient inside any compact set $U$ for some finite $L$, and $\sup_{\theta \in U}\|\nabla^2 f(\theta)\| \leq L$ by Lemma 0. Now define $U_\alpha = \{\theta - t\alpha\nabla f(\theta) \mid t \in [0, 1], \theta \in U_0\}$ and the continuous function $L(\alpha) = \sup_{\theta \in U_\alpha}\|\nabla^2 f(\theta)\|$. Notice that $U_0 \subset U_{\alpha'}$ for all $\alpha$. We prove that $\alpha L(\alpha) < 2$ implies $U_\alpha = U_0$ and in particular, $L(\alpha) = L(0)$. By Taylor expansion,

$$f(\theta - t\alpha\nabla f) = f(\theta) - \alpha\|\nabla f(\theta)\|^2 + \frac{t^2\alpha^2}{2}\nabla f(\theta)^T\nabla^2 f(\theta - t'\alpha\nabla f)f(\theta)$$

for some $t' \in [0, t] \subset [0, 1]$. Since $\theta - t'\alpha\nabla f \in U_\alpha$, it follows that

$$f(\theta - t\alpha\nabla f) \leq f(\theta) - \alpha\|\nabla f(\theta)\|^2(1 - \alpha L(\alpha)/2) \leq f(\theta)$$

for all $\alpha L(\alpha) < 2$. In particular, $\theta - t\alpha\nabla f \in U_0$ and hence $U_\alpha = U_0$. We conclude that $\alpha L(\alpha) < 2$ implies $L(\alpha) = L(0)$, implying in turn $\alpha L(0) < 2$. We now claim the converse, namely that $\alpha L(0) < 2$ implies $\alpha L(\alpha) < 2$. For contradiction, assume otherwise that there exists $\alpha'L(0) < 2$ with $\alpha'L(\alpha') \geq 2$. Since $\alpha L(\alpha)$ is continuous and $0L(0) = 0 < 2$, there exists $\bar{\alpha} \leq \alpha'$ such that $\bar{\alpha}L(0) < 2$ and $\bar{\alpha}L(\bar{\alpha}) = 2$. This is in contradiction with continuity:

$$2 = \bar{\alpha}L(\bar{\alpha}) = \lim_{\alpha \to \bar{\alpha}^-}\alpha L(\alpha) = \lim_{\alpha \to \bar{\alpha}^-}\alpha L(0) = \bar{\alpha}L(0).$$

Finally we conclude that $U_\alpha = U_0$ for all $\alpha L(0) < 2$, and in particular, for all $\alpha L < 2$. Finally, $\theta_k \in U_0$ implies $\theta_{k+1} \in U_\alpha = U_0$ and hence $\theta_k \in U_0$ by induction. The result now follows by applying the previous proposition to $f|_{U_0}$. $\qquad\square$

## C  PROOF OF PROPOSITION 2

**Proposition 2.** *Write $\lambda(A) = \mathrm{Re}(\mathrm{Spec}(A))$ for real parts of the eigenvalues of a matrix $A$. We have the following implications, and none of them are equivalences.*

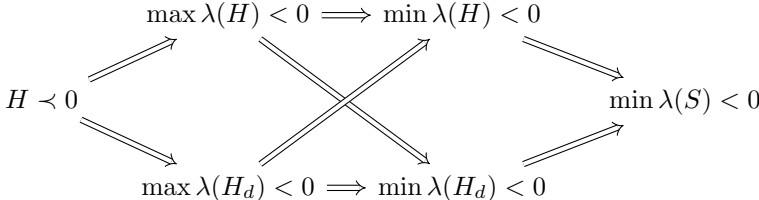

The top row is dynamics-based, governed by the collective Hessian, while the bottom row is game-theoretic whereby $H_d = \bigoplus \nabla_{ii}L^i$ decomposes into agentwise Hessians. The left and right triangles collide respectively to strict maxima and saddles for single losses, since $H = S = H_d = \nabla^2 L$.

*Proof.* First note that $H \prec 0 \iff S \prec 0 \iff \max\lambda(S) < 0$, so the leftmost term can be replaced by $\max\lambda(S) < 0$.

We begin with the leftmost implications. If $\max\lambda(S) < 0$ then $S \prec 0$ by symmetry of $S$, implying both $H \prec 0$ since $u^T H u = u^T S u$ for all $u \in \mathbb{R}^d$, and negative definite diagonal blocks $\nabla^2 L^i i \prec 0$; finally $H_d \prec 0$. In particular this implies $\max\lambda(H) < 0$ and $\max\lambda(H_d) \prec 0$ since real parts of eigenvalues of a negative definite matrix are negative.

The rightmost implications follow as above by contraposition: if $\min\lambda(S) \geq 0$ then $S \succeq 0$, which implies $H \succeq 0$ and $H_d \succeq 0$ and hence $\min\lambda(H) \geq 0$, $\min\lambda(H_d) \geq 0$.

The top and bottom implications are trivial.

The diagonal implications hold by a trace argument:

$$\sum_i \lambda_i(H) = Tr(H) = \mathrm{Tr}(H_d) = \sum_i \lambda_i(H_d),$$

hence $\max \lambda(H) < 0$ implies the LHS is negative and thus $\sum_i \lambda_i(H_d) < 0$. It follows that $\lambda_i(H_d) < 0$ for some $i$ and finally $\min \lambda(H_d) < 0$. The other diagonal holds identically.

We now prove that no implication is an equivalence. For the leftmost implications,

$$H = \begin{pmatrix} -1 & 2 \\ 2 & -1 \end{pmatrix}$$

has $\max \lambda(H_d) = -1 < 0$ while $\max \lambda(S) = 3 > 0$, and

$$H = \begin{pmatrix} 2 & 4 \\ -4 & -4 \end{pmatrix}$$

has $\max \lambda(H) = -1 < 0$ while $\max \lambda(S) = 2 > 0$. This also proves the diagonal implications: the first matrix has $\min \lambda(H_d) = -1 < 0$ but $\max \lambda(H) = 3 > 0$, and the second matrix has $\min \lambda(H) = -1 < 0$ but $\max \lambda(H_d) = 2 > 0$.

For the rightmost implications, swap the sign of the diagonal elements for the two matrices above.

The top and bottom implications are trivially not equivalences:

$$H = H_d = \begin{pmatrix} 1 & 0 \\ 0 & -1 \end{pmatrix}$$

has $\min \lambda(H) = \min \lambda(H_d) = -1 < 0$ but $\max \lambda(H) = \max \lambda(H_d) = 1 > 0$. $\qquad\square$

## D  PROOF OF THEOREM 1

The variable changes

$$(x', y') = (y, -x), \qquad (x', y') = (-y, x), \qquad (x', y') = (-x, -y) \qquad (\dagger)$$

will be useful, taking the positive quadrant $x, y \geq 0$ to the other three.

**Theorem 1.** *There is a coercive, nondegenerate, analytic two-player market $\mathcal{M}$ whose only critical point is a strict maximum. In particular, algorithms only have four possible outcomes in $\mathcal{M}$:*

1. *Iterates are unbounded, and all players diverge to infinite loss. [Not global]*

2. *Iterates are bounded and converge to the strict maximum. [Not reasonable]*

3. *Iterates are bounded and converge to a non-critical point. [Not reasonable]*

4. *Iterates are bounded but do not converge (cycle). [Not global]*

For intuition purposes, $\mathcal{M}$ was constructed by noticing that there is no necessary reason for the local minima of two coercive losses to coincide: the gradients of each loss may only *simultaneously* vanish at a local maximum in each player's respective coordinate. The highest-order terms (first and last) provide coercivity in both coordinates while still having zero-sum interactions. The $-x^2$ and $-y^2$ terms yield a strict local maximum *at* the origin, while the $\pm xy$ terms provide opposite incentives *around* the origin, preventing any other simultaneous critical point to arise.

*Proof.* Write $\theta = (x, y)$ and consider the analytic market $\mathcal{M}$ given by

$$L^1 = x^6/6 - x^2/2 + xy + \frac{1}{4}\left(\frac{y^4}{1 + x^2} - \frac{x^4}{1 + y^2}\right)$$

$$L^2 = y^6/6 - y^2/2 - xy - \frac{1}{4}\left(\frac{y^4}{1 + x^2} - \frac{x^4}{1 + y^2}\right)$$

with simultaneous gradient

$$
\xi = \begin{pmatrix} x^5 - x + y - \frac{y^4 x}{2(1+x^2)^2} - \frac{x^3}{1+y^2} \\ y^5 - y - x - \frac{x^4 y}{2(1+y^2)^2} - \frac{y^3}{1+x^2} \end{pmatrix}.
$$

We prove 'by hand' that the origin $\bar{\theta} = 0$ is the only critical point (solution to $\xi = 0$). See further down for an easier approach based on Sturm's theorem, computer-assisted though equally rigorous.

We can assume $x, y \geq 0$ since any other solution can be obtained by a quadrant variable change (†). Now assume for contradiction that $\xi = 0$ with $y \neq 0$.

**1.** We first show that $y > 1$. Indeed,

$$
0 = \xi_2 = y^5 - y - x - \frac{x^4 y}{2(1+y^2)^2} - \frac{y^3}{1+x^2} < y^5 - y = y(y^4 - 1)
$$

implies $y > 1$ since $y \geq 0$.

**2.** We now show that $y < 1.5$. First assume for contradiction that $x \geq y$, then

$$
\xi_1 = y - x + x^5 - \frac{xy^4}{2(1+x^2)^2} - \frac{x^3}{1+y^2} > 1 - x + x^5 - x^5/8 - x^3/2 := h(x).
$$

Now

$$
h'(x) = \frac{35}{8} x^4 - \frac{3}{2} x^2 - 1
$$

has unique positive root

$$
x_0 = \sqrt{\frac{6 + 2\sqrt{79}}{35}}
$$

and $h(x) \to \infty$ as $x \to \infty$, hence $h$ attains its minimum at $x_0$ and plugging $x_0$ yields a contradiction

$$
\xi_1 > h(x_0) > 0.
$$

We conclude that $x < y$, but combining this with $x \geq 0$ yields

$$
\xi_2 > -2y + y^5 - y^5/8 - y^3 = y(7y^4/8 - y^2 - 2) > 7y^4/8 - y^2 - 2 > 0
$$

for all $y \geq 1.5$, since the rightmost polynomial is positive at $y = 1.5$ and has positive derivative

$$
7y^3/2 - 2y = y(7y^2/2 - 2) \geq 7(1.5)^2/2 - 2 > 0.
$$

We must therefore have $y < 1.5$ as required.

**3.** It remains only to show that $\xi_1 > 0$ for all $1 < y < 1.5$. First notice that $f_x(y) = \xi_1(x, y)$ is concave in $y$ for any fixed $x \geq 0$ since

$$
f'_x(y) = 1 - \frac{2y^3 x}{(1+x^2)^2} + 2x^3 \frac{y}{(1+y^2)^2}
$$

and so

$$
f''_x(y) = -\frac{6y^2 x}{(1+x^2)^2} + 2x^3 \frac{1 + y^2 - 4y^2}{(1+y^2)^3} = -\frac{6y^2 x}{(1+x^2)^2} - 2x^3 \frac{3y^2 - 1}{(1+y^2)^3} \leq 0
$$

for $y > 1$. It follows that $f_x$ attains its infimum on the boundary $y \in \{1, 1.5\}$, so it suffices to check that $\xi_1(x, 1) > 0$ and $\xi_1(x, 1.5) > 0$ for all $x \geq 0$. First notice that

$$
g(x) := \frac{x}{2(1+x^2)^2}
$$

satisfies

$$
g'(x) = \frac{1 + x^2 - 4x^2}{2(1+x^2)^2} = \frac{1 - 3x^2}{2(1+x^2)^2},
$$

which has a unique positive root at $x_0 = 1/\sqrt{3}$. This critical point of $g$ must be a maximum since $g(x) > 0$ for $x > 0$ and $g(x) \to 0$ as $x \to \infty$. It follows that

$$g(x) \leq g(x_0) = \frac{1}{2\sqrt{3}(1 + 1/3)^2} = 3\sqrt{3}/32 \,.$$

We now obtain

$$\xi_1(x, 1) \geq x^5 - x^3/2 - x + 1 - 3\sqrt{3}/32 \coloneqq p(x)$$

and

$$\xi_1(x, 1.5) \geq x^5 - 4x^3/13 - x + 1.5 - (1.5)^4 3\sqrt{3}/32 \coloneqq q(x) \,.$$

Notice that

$$p'(x) = 5x^4 - 3x^2/2 - 1$$

has unique positive root

$$x_0 = \sqrt{\frac{3 + \sqrt{89}}{20}}$$

and $p(x) \to \infty$ as $x \to \infty$, hence $p$ attains its minimum at $x_0$ and plugging $x_0$ yields

$$\xi_1(x, 1) \geq p(x_0) > 0 \,.$$

Similarly for $q$ we have

$$q'(x) = 5x^4 - 12x^2/13 - 1$$

has unique positive root

$$x_0 = \sqrt{\frac{6 + \sqrt{881}}{65}}$$

and plugging $x_0$ yields

$$\xi_1(x, 1.5) \geq q(x_0) > 0 \,.$$

We conclude that

$$\xi_1(x, y) \geq \min(\xi_1(x, 1), \xi_1(x, 1.5)) > 0$$

and the contradiction is complete, hence $y = 0$. Finally $\xi_2 = 0 = x$, so $\bar{\theta} = 0$ is the unique critical point as required. Now the Hessian at $\bar{\theta}$ is

$$H(\bar{\theta}) = \begin{pmatrix} -1 & 1 \\ -1 & -1 \end{pmatrix} \,,$$

which is negative definite since $S(\bar{\theta}) = -I \prec 0$, so $\bar{\theta}$ is a nondegenerate strict maximum and $\mathcal{M}$ is nondegenerate. It remains only to prove coercivity of $\mathcal{M}$, namely coercivity of $L^1$ and $L^2$. Coercivity of $L^1$ follows by noticing that the dominant terms are $x^6/6$ and $y^4/(1 + x^2)$. Formally, first note that $\frac{x^4}{1+y^2} \leq x^4$, hence

$$L^1 \geq x^6/6 - x^4/4 - x^2/2 + xy + \frac{1}{4}\left(\frac{y^4}{1 + x^2}\right) \,.$$

Now $xy \geq -|xy| \geq -(2x^2 + y^2/8)$ by Young's inequality, hence

$$L^1 \geq x^6/6 - x^4/4 - 5x^2/2 - y^2/8 + \frac{1}{4}\left(\frac{y^4}{1 + x^2}\right) \,.$$

For any sequence $\|\theta\| \to \infty$, either $|x| \to \infty$ or $|x|$ is bounded above by some $k \in \mathbb{R}$ and $|y| \to \infty$. In the latter case, we have

$$\lim_{\|\theta\| \to \infty} L^1 \geq \lim_{|y| \to \infty} -k^4/4 - 5k^2/2 - y^2/8 + \frac{y^4}{4(1 + k^2)} = \infty$$

since the leading term $y^4$ is of even degree and has positive coefficient, so we are done. Otherwise, for $|x| \to \infty$, we pursue the previous inequality to obtain

$$L^1 \geq x^6/6 - x^4/4 - 5x^2/2 + \frac{y^2}{8}\left(\frac{2y^2}{1 + x^2} - 1\right) \,.$$

Now notice that $y^2 \geq x^2 \geq 1$ implies

$$L^1 \geq x^6/6 - x^4/4 - 5x^2/2 + \frac{x^2}{8}\left(\frac{x^2-1}{1+x^2}\right) \geq x^6/6 - x^4/4 - 5x^2/2 - x^2/8\,.$$

On the other hand, $x^2 \geq y^2$ also implies

$$L^1 \geq x^6/6 - x^4/4 - 5x^2/2 - x^2/8$$

by discarding the first (positive) term in the brackets. Both cases lead to the same inequality and hence, for any sequence with $|x| \to \infty$,

$$\lim_{\|\theta\| \to \infty} L^1 \geq \lim_{|x| \to \infty} x^6/6 - x^4/4 - 5x^2/2 - x^2/8 = \infty$$

since the leading term $x^6$ has even degree and positive coefficient. Hence $L^1$ is coercive, and the same argument holds for $L^2$ by swapping $x$ and $y$. As required we have constructed a coercive, nondegenerate, analytic two-player market $\mathcal{M}$ whose only critical point is a strict maximum.

In particular, any algorithm either has unbounded iterates with infinite losses or bounded iterates. If they are bounded, they either fail to converge or converge. If they converge, they either converge to a non-critical point or a critical point, which can only be the strict maximum.

[For an alternative proof that $\bar{\theta} = 0$ is the only critical point, we may take advantage of computer algebra systems to find the exact number of real roots using the resultant matrix and Sturm's theorem. Singular (Decker et al., 2019) is one such free and open-source system for polynomial computations, backed by published computer algebra references. In particular, the `rootsur` library used below is based on the book by Basu et al. (2006). First convert the equations into polynomials:

$$\begin{cases} 2(1+x^2)^2(1+y^2)(x^5-x+y) - y^4x(1+y^2) - 2x^3(1+x^2)^2 = 0 \\ 2(1+y^2)^2(1+x^2)(y^5-y-x) - x^4y(1+x^2) - 2y^3(1+y^2)^2 = 0\,. \end{cases}$$

We compute the resultant matrix determinant of the system with respect to $y$, a univariate polynomial $P$ in $x$ whose zeros are guaranteed to contain all solutions in $x$ of the initial system. We then use the Sturm sequence of $P$ to find its exact number of real roots. This is implemented with the Singular code below, whose output is 1.

```
LIB "solve.lib"; LIB "rootsur.lib";
ring r = (0,x),(y),dp;
poly p1 = 2*(1+x^2)^2*(1+y^2)*(x^5-x+y)-y^4*x*(1+y^2)-2*x^3*(1+x^2)^2;
poly p2 = 2*(1+y^2)^2*(1+x^2)*(y^5-y-x)-x^4*y*(1+x^2)-2*y^3*(1+y^2)^2;
ideal i = p1,p2;
poly f = det(mp_res_mat(i));
ring s = 0,(x,y),dp; poly f = imap(r, f);
nrroots(f);
```

We know that $\bar{\theta} = 0$ is a real solution, so $\bar{\theta}$ must be the unique critical point.]  □

## E    PROOF OF THEOREM 2

**Theorem 2.** *Given a reasonable algorithm with bounded continuous distribution on $\theta_0$ and a real number $\epsilon > 0$, there exists a coercive, nondegenerate, almost-everywhere analytic two-player market $\mathcal{M}_\sigma$ with a strict minimum and no other critical points, such that $\theta_k$ either cycles or diverges to infinite losses for both players with probability at least $1 - \epsilon$.*

*Proof.* We modify the construction from Theorem 1 by deforming a small region around the maximum to replace it with a minimum. First let $0 < \sigma < 0.1$ and define

$$f_\sigma(\theta) = \begin{cases} (x^2 + y^2 - \sigma^2)/2 & \text{if } \|\theta\| \geq \sigma \\ (y^2 - 3x^2)(x^2 + y^2 - \sigma^2)/(2\sigma^2) & \text{otherwise,} \end{cases}$$

where $\theta = (x, y)$ and $\|\theta\| = \sqrt{x^2 + y^2}$ is the standard $L2$-norm. Note that $f_\sigma$ is continuous since

$$\lim_{\|\theta\| \to \sigma^+} f_\sigma(\theta) = 0 = \lim_{\|\theta\| \to \sigma^-} f_\sigma(\theta)\,.$$

Now consider the two-player market $\mathcal{M}_\sigma$ given by

$$L^1 = x^6/6 - x^2 + f_\sigma + xy + \frac{1}{4}\left(\frac{y^4}{1+x^2} - \frac{x^4}{1+y^2}\right)$$

$$L^2 = y^6/6 - f_\sigma - xy - \frac{1}{4}\left(\frac{y^4}{1+x^2} - \frac{x^4}{1+y^2}\right).$$

The resulting losses are continuous but not differentiable; however, they are analytic (in particular smooth) almost everywhere, namely, for all $\theta$ not on the circle of radius $\sigma$. This is sufficient for the purposes of gradient-based optimization, noting that neural nets also fail to be everywhere-differentiable in the presence of rectified linear units.

We claim that $\mathcal{M}_\sigma$ has a single critical point at the origin $\bar\theta = 0$. First note that

$$\xi_{\mathcal{M}_\sigma} = \xi_{\mathcal{M}_0} = \begin{pmatrix} x^5 - x + y - \frac{y^4 x}{2(1+x^2)^2} - \frac{x^3}{1+y^2} \\ y^5 - y - x - \frac{x^4 y}{2(1+y^2)^2} - \frac{y^3}{1+x^2} \end{pmatrix} = \xi_{\mathcal{M}}$$

for all $\|\theta\| \geq \sigma$, where $\mathcal{M}$ is the game from Theorem 1. It was proved there that the only real solution to $\xi = 0$ is the origin, which does not satisfy $\|\theta\| \geq \sigma$. Any critical point must therefore satisfy $\|\theta\| < \sigma$, for which

$$\xi = \xi_{\mathcal{M}_\sigma} = \begin{pmatrix} x^5 + x + y - 2x(3x^2 + y^2)/\sigma^2 - \frac{y^4 x}{2(1+x^2)^2} - \frac{x^3}{1+y^2} \\ y^5 + y - x - 2y(y^2 - x^2)/\sigma^2 - \frac{x^4 y}{2(1+y^2)^2} - \frac{y^3}{1+x^2} \end{pmatrix}.$$

First note that $\bar\theta = 0$ is a critical point; we prove that there are no others. The continuous parameter $\sigma$ prevents us from using a formal verification system, so we must work 'by hand'. Warning: the proof is a long inelegant string of case-by-case inequalities.

Assume for contradiction that $\xi = 0$ with $\theta \neq 0$. First note that $\|\theta\| < \sigma$ implies $|x|, |y| < \sigma$, and $x = 0$ or $y = 0$ implies $x = y = 0$ using $\xi_1 = 0$ or $\xi_2 = 0$ respectively. We can therefore assume $0 < |x|, |y| < \sigma$. We can moreover assume that $x > 0$, the opposite case following by the quadrant change of variables $(x', y') = (-x, -y)$.

1. We begin with the case $\sigma/2 \leq x < \sigma$. First notice that

$$x + y - 2x(3x^2 + y^2)/\sigma^2 = x(1 - 6x^2/\sigma^2) + y(1 - 2xy/\sigma^2) \leq x(1 - 3/2) + y(1 - y/\sigma)$$

and the rightmost term attains its maximum value for $y = \sigma/2$, hence

$$x + y - 2x(3x^2 + y^2)/\sigma^2 \leq -x/2 + \sigma/4 \leq 0.$$

This implies

$$\xi_1 \leq x^5 - \frac{y^4 x}{2(1+x^2)^2} - \frac{x^3}{1+y^2} < x^5 - \frac{x^3}{1+y^2} < x^3\left(1 - y^2 - \frac{1}{1+y^2}\right) = \frac{-x^3 y^4}{1+y^2} < 0$$

using $x^2 + y^2 < 1$, which is a contradiction to $\xi = 0$.

2. We proceed with the case $x < \sigma/2$ and $|y| \leq \sigma/2$. First, $y < 0$ implies the contradiction

$$\xi_2 < y - 2y^3/\sigma^2 - \frac{x^4 y}{2(1+y^2)^2} - \frac{y^3}{1+x^2} < y/2 - y\left(\frac{\sigma^4}{2^5} + \frac{\sigma^2}{2^2}\right) < y\left(\frac{1}{2} - \frac{1}{2^5} - \frac{1}{2^2}\right) < 0,$$

so we can assume $y > 0$. In particular we have $(1 - 2y(y + x)/\sigma^2) > 0$. If $y \leq x$, we also obtain

$$\xi_2 < y^5 + (y - x)\left(1 - 2y(y + x)/\sigma^2\right) - \frac{y^3}{1+x^2} < y^3\left(y^2 - \frac{1}{1+x^2}\right) < \frac{-y^3 x^4}{1+x^2} < 0,$$

so we can assume $x < y$. There are again two cases to distinguish. If $x < \sigma/2 - b\sigma^2$ with $b = 0.08$,

$$x(1 - 6x^2/\sigma^2) + y(1 - 2xy/\sigma^2) > x(1 - 3(1/2 - \sigma b)) + x(1 - (1/2 - \sigma b)) > 4\sigma bx$$

which implies the contradiction

$$\xi_1 > 4\sigma bx - \frac{y^4 x}{2(1+x^2)^2} - \frac{x^3}{1+y^2} > \sigma x\left(4b - \frac{\sigma^4}{2^5} - \frac{\sigma^2}{2^2}\right) > \sigma x\left(4b - \frac{1}{2^5} - \frac{1}{2^2}\right) > 0\,.$$

Finally assume $x \geq \sigma/2 - b\sigma^2$. Then we have

$$(y-x)(1 - 2y(x+y)/\sigma^2) < b\sigma^2(1 - 4x^2/\sigma^2) < b\sigma^2(1 - (1-2\sigma b)^2) = 4\sigma^3 b^2(1-\sigma b) < 4\sigma^3 b^2$$

and obtain

$$\xi_2 < y^5 + 4\sigma^3 b^2 - \frac{y^3}{1+x^2} < \sigma^3\left(\sigma^2/2^5 + 4b^2 - \frac{(1/2 - \sigma b)^3}{1 + \sigma^2/4}\right)\,.$$

We claim that the rightmost term is negative. Indeed, the quantity inside the brackets has derivative

$$\sigma/2^4 + \frac{(1/2 - \sigma b)^2}{(1 + \sigma^2/4)^2}\left(3b(1 + \sigma^2/4) + \sigma(1/2 - \sigma b)/2\right) > 0$$

and so its supremum across $\sigma \in [0, 0.1]$ must be attained at $\sigma = 0.1$. We obtain the contradiction

$$\xi_2 < \sigma^3\left(0.01/2^5 + 4b^2 - \frac{(1/2 - b)^3}{1 + 0.01/4}\right) < 0$$

for $b = 0.08$ and $\sigma > 0$, as required.

3. Finally, consider the case $x < \sigma/2$ and $|y| > \sigma/2$. First, $y < 0$ implies the contradiction

$$\xi_1 < x + y - 2x(3x^2 + y^2)/\sigma^2 < -2x(3x^2 + y^2) < 0$$

so we can assume $y > 0$. Now assume $y < \sigma - x(1 + \sigma^2)$. Then

$$x(1 - 6x^2/\sigma^2) + y(1 - 2xy/\sigma^2) > -x/2 + y(1 - y/\sigma) > -x/2 + x(1 + \sigma^2) > x(1/2 + \sigma^2)\,,$$

which yields the contradiction

$$\xi_1 > x\left(\frac{1}{2} + \sigma^2 - \frac{y^4}{2(1+x^2)^2} - \frac{x^2}{1+y^2}\right) > x\left(1/2 + \sigma^2 - \sigma^4 - \sigma^2/4\right) > x(1/2 - 1/4) > 0\,.$$

We can therefore assume $y \geq \sigma - x(1 + \sigma^2)$. We have

$$(y - x)(1 - 2y(y+x)/\sigma^2) < (y-x)(1 - (y+x)/\sigma) \leq (y-x)(1 - (1 - \sigma x)) < \sigma x(y-x)$$

which attains its maximum in $x$ at $x = y/2$, hence

$$\xi_2 < y^5 - \frac{y^3}{1+x^2} + \frac{\sigma y^2}{4} < \frac{\sigma y^2}{4}\left(4\sigma^2 - \frac{2}{1+\sigma^2} + 4\right)\,.$$

Finally we obtain the contradiction

$$\xi_2 < \frac{\sigma y^2}{4}\left(\frac{5\sigma^2 + 4\sigma^4 - 1}{1 + \sigma^2}\right) < 0$$

for all $\sigma < 0.1$. All cases lead to contradictions, so we conclude that $\bar\theta$ is the only critical point, with positive definite Hessian

$$H(\bar\theta) = \begin{pmatrix} 1 & 1 \\ -1 & 1 \end{pmatrix} \succ 0\,,$$

hence $\bar\theta$ is a strict minimum. Now notice that $\mathcal{M}_0$ has the same dominant terms as $\mathcal{M}$ from Theorem 1, so coercivity of $\mathcal{M}_0$ follows from the same argument. Since $\mathcal{M}_\sigma$ is identical to $\mathcal{M}_0$ outside the $\sigma$-ball $B_\sigma = \{(x, y) \in \mathbb{R}^2 \mid \|\theta\| < \sigma\}$, coercivity of $\mathcal{M}_0$ implies coercivity of $\mathcal{M}_\sigma$ for any $\sigma$.

Fix any reasonable algorithm $F$, any bounded continuous measure $\nu$ on $\mathbb{R}^d$ with initial region $U$, and any $\epsilon > 0$. We abuse notation somewhat and write $F_\sigma^k(\theta_0)$ for the $k$th iterate of $F$ in $\mathcal{M}_\sigma$ with initial parameters $\theta_0$. We claim that there exists $\sigma > 0$ such that

$$P_\nu\left(\theta_0 \in U \text{ and } \lim_k F_\sigma^k(\theta_0) = \bar\theta\right) < \epsilon\,.$$

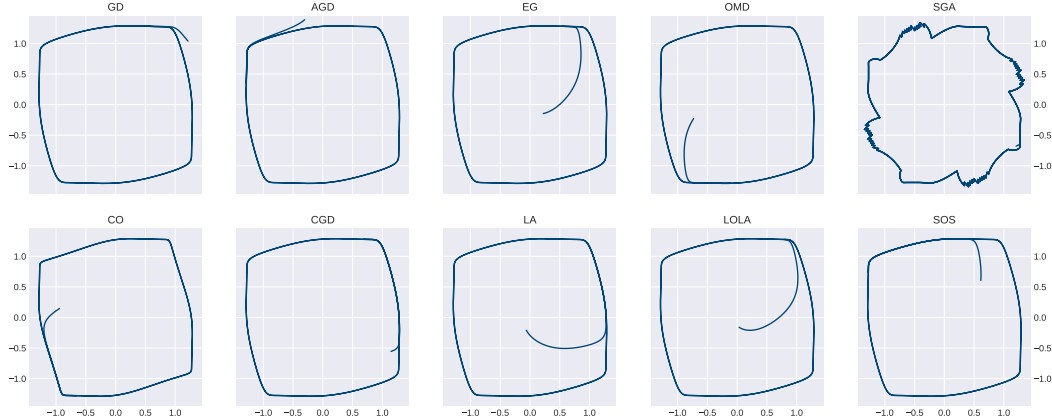

Figure 2: Algorithms in $\mathcal{A}$ fail to converge in $\mathcal{M}_\sigma$ with $\sigma = \alpha = \gamma = 0.01$. Single run with standard normal initialisation, 3000 iterations.

Since $\bar{\theta}$ is the only critical point and $\mathcal{M}_\sigma$ is coercive, this implies bounded but non-convergent iterates or divergent iterates with infinite losses with probability at least $1 - \epsilon$, proving the theorem. To begin, $\mu(B_\sigma) \to 0$ as $\sigma \to 0$ implies that we can pick $\sigma' > 0$ such that $P_\nu(\theta_0 \in B_{\sigma'}) < \epsilon/2$ by continuity of $\nu$ with respect to Lebesgue measure.

Now let $\bar{U}$ be the closure of $U$ and define $D = \bar{U} \cap \{\|\theta\| \geq \sigma'\}$. Note that $D$ is compact since $\bar{U}$ is compact and closed subsets of a compact set are compact. $F$ is reasonable, $D$ is bounded and $\bar{\theta} = 0$ is a strict maximum in $\mathcal{M}_0$, so there are hyperparameters such that the stable set

$$Z = \{\theta_0 \in D \mid \lim_k F_0^k(\theta_0) = 0\}$$

has zero measure. We claim that

$$Z_\delta := \{\theta_0 \in D \mid \inf_{k \in \mathbb{N}} \left\| F_0^k(\theta_0) \right\| < \delta\}$$

has arbitrarily small measure as $\delta \to 0$. Assume for contradiction that there exists $\alpha > 0$ such that $\mu(Z_\delta) \geq \alpha$ for all $\delta > 0$. Then $Z_\delta \subset Z_{\delta'}$ and $\mu(Z_\delta) \leq \mu(D) < \infty$ for all $\delta < \delta'$ implies

$$\mu\left(\bigcap_{n \in \mathbb{N}} Z_{\frac{1}{n}}\right) = \lim_{n \to \infty} \mu\left(Z_{\frac{1}{n}}\right) \geq \alpha$$

by Nelson (2015, Exercise 1.19). On the other hand,

$$\bigcap_{n \in \mathbb{N}} Z_{\frac{1}{n}} = Z_0$$

yields the contradiction $0 = \mu(Z_0) \geq \alpha$. We conclude that $Z_\delta$ has arbitrarily small measure, hence there exists $\delta > 0$ such that

$$P_\nu(\theta_0 \in Z_\delta) < \epsilon/2$$

by continuity of $\nu$. Now let $\sigma = \min\{\sigma', \delta\}$ and notice that

$$\theta_0 \in D \setminus Z_\delta \quad \implies \quad \inf_k \left\| F_0^k(\theta_0) \right\| \geq \delta \geq \sigma \quad \implies \quad \inf_k \left\| F_\sigma^k(\theta_0) \right\| \geq \sigma,$$

where the last implication holds since $\mathcal{M}_\sigma$ and $\mathcal{M}_0$ are indistinguishable in $\{\|\theta\| \geq \sigma\}$, so the algorithm must have identical iterates $F_\sigma^k(\theta_0) = F_0^k(\theta_0)$ for all $k$. It follows by contraposition that $\lim_k F_\sigma^k(\theta_0) = \bar{\theta}$ implies $\inf_k \left\| F_\sigma^k(\theta_0) \right\| < \sigma$ and so $\theta_0 \in Z_\delta$ or $\theta_0 \notin D$. Finally we obtain

$$P_\nu\left(\theta_0 \in U \text{ and } \lim_k F_\sigma^k(\theta_0) = \bar{\theta}\right) = P_\nu(\theta_0 \in U \cap Z_\delta \text{ or } \theta_0 \in U \setminus D)$$

$$\leq P_\nu(\theta_0 \in U \cap Z_\delta) + P_\nu(\theta_0 \in U \setminus D)$$

$$\leq P_\nu(\theta_0 \in Z_\delta) + P_\nu(\theta_0 \in B_{\sigma'})$$

$$< \epsilon/2 + \epsilon/2 = \epsilon$$

as required. We plot iterates for a single run of each algorithm in Figure 3 with $\alpha = \gamma = 0.01$. $\quad \square$

# F  PROOF OF THEOREM 3

**Theorem 3.** *There is a weakly-coercive, nondegenerate, analytic two-player zero-sum game $\mathcal{N}$ whose only critical point is a strict maximum. Algorithms in $\mathcal{A}$ almost surely have bounded non-convergent iterates in $\mathcal{N}$ for $\alpha, \gamma$ sufficiently small.*

*Proof.* Consider the analytic zero-sum game $\mathcal{N}$ given by

$$L^1 = xy - x^2/2 + y^2/2 + x^4/4 - y^4/4 = -L^2$$

with simultaneous gradient

$$\xi = \begin{pmatrix} y - x + x^3 \\ -x - y + y^3 \end{pmatrix}$$

and Hessian

$$H = \begin{pmatrix} -1 + 3x^2 & 1 \\ -1 & -1 + 3y^2 \end{pmatrix}.$$

We show that the only solution to $\xi = 0$ is the origin. First we can assume $x, y \geq 0$ since any other solution can be obtained by a quadrant variable change (†). Now assume for contradiction that $y \neq 0$, then

$$\xi_2 = 0 = -x - y + y^3 \leq -y + y^3 = y(y^2 - 1)$$

implies $y \geq 1$ and hence

$$\xi_1 = 0 = y - x + x^3 \geq 1 - x + x^3 = (x + 1)(x - 1)^2 + x^2 > 0$$

which is a contradiction. It follows that $y = 0$ and hence $\xi_2 = 0 = x$ as required. Now the origin has invertible, negative-definite Hessian

$$H(0) = \begin{pmatrix} -1 & 1 \\ -1 & -1 \end{pmatrix} \prec 0$$

so the unique critical point is a strict maximum. The game is nondegenerate since the only critical point has invertible Hessian. The game is weakly-coercive since $L^1(x, \bar{y}) \to \infty$ for any fixed $\bar{y}$ by domination of the $x^4$ term; similarly for $L^2(\bar{x}, y)$ by domination of the $y^4$ term.

**Bounded iterates: strategy.**  We begin by showing that all algorithms have bounded iterates in $\mathcal{N}$ for $\alpha, \gamma$ sufficiently small. For each algorithm $F$, our strategy is to show that there exists $r > 0$ such that for any $s > 0$ we have $\|F(\theta)\| < \|\theta\|$ for all $r < \|\theta\| < s$ and $\alpha, \gamma$ sufficiently small. This will be enough to prove bounded iteration upon bounded initialisation. Denote by $B_r$ the ball of radius $r$ centered at the origin.

**GD.**  We have

$$\begin{aligned}
\theta^T \xi &= x(y - x + x^3) + y(-x - y + y^3) \\
&= x^4 - x^2 + y^4 - y^2 \\
&= (x^2 - 1)^2 + (y^2 - 1)^2 + x^2 + y^2 - 2 > 1
\end{aligned}$$

for all $\|\theta\|^2 = x^2 + y^2 > 3$. For any $s > 0$ we obtain

$$\|F(\theta)\|^2 = \|\theta - \alpha\xi\|^2 = \|\theta\|^2 - 2\alpha\theta^T\xi + \alpha^2 \|\xi\|^2 < \|\theta\| - \alpha\left(2 - \alpha \|\xi\|^2\right) < \|\theta\|^2$$

for all $\sqrt{3} < \|\theta\| < s$ and $\alpha$ sufficiently small, namely $0 < \alpha < 2/\sup_{\theta \in B_s} \|\xi\|^2$.

**EG.**  For any $s > 0$ and $\sqrt{4} < \|\theta\| < s$ we have

$$\|\theta - \alpha\xi(\theta)\|^2 > 4 - 2\alpha\theta^T\xi > 3$$

for $\alpha < 1/\sup_{\theta \in B_s} 2\theta^T\xi$. Now using $\theta^T\xi > 1$ for all $\|\theta\|^2 > 3$ by the argument for GD above,

$$\begin{aligned}
\|F(\theta)\|^2 &= \|\theta\|^2 - 2\alpha\theta^T\xi(\theta - \alpha\xi(\theta)) + \alpha^2 \|\xi(\theta - \alpha\xi(\theta))\|^2 \\
&= \|\theta\|^2 - 2\alpha(\theta - \alpha\xi(\theta))^T\xi(\theta - \alpha\xi(\theta)) + O(\alpha^2) \\
&< \|\theta\|^2 - \alpha\left(2 - O(\alpha)\right) < \|\theta\|^2
\end{aligned}$$

for $\alpha$ sufficiently small.

**AGD.** For any $s > 0$, notice by continuity of $\xi$ that there exists $\delta > 0$ such that

$$\theta^T(\xi_1, \xi_2(\theta_1 - \alpha\xi_1, \theta_2)) > \theta^T\xi - 1/2$$

for all $\alpha < \delta$ and $\theta \in B_s$, since $B_s$ is bounded and $\theta_1 - \alpha\xi_1 \to \theta_1$ as $\alpha \to 0$. It follows that

$$\begin{aligned}
\|F(\theta)\|^2 &= \|\theta\|^2 - 2\alpha\theta^T(\xi_1, \xi_2(\theta_1 - \alpha\xi_1, \theta_2)) + O(\alpha^2) \\
&< \|\theta\|^2 - 2\alpha(\theta^T\xi - 1/2) + O(\alpha^2) \\
&< \|\theta\|^2 - 2\alpha(1 - 1/2) + O(\alpha^2) \\
&< \|\theta\|^2 - \alpha(1 - O(\alpha)) < \|\theta\|^2
\end{aligned}$$

for all $\sqrt{3} < \|\theta\| < s$ and $\alpha < \delta$ sufficiently small.

**OMD.** For any $s > 0$, notice by continuity of $\xi$ that there exists $\delta > 0$ such that

$$\left|\theta^T(\xi(\theta) - \xi((\mathrm{id} - \alpha\xi)^{-1}(\theta)))\right| < 1/2$$

for all $\alpha < \delta$ and $\theta \in B_s$, since $B_s$ is bounded and $(\mathrm{id} - \alpha\xi)^{-1}(\theta) \to \theta$ as $\alpha \to 0$. It follows that

$$\begin{aligned}
\|F(\theta)\|^2 &= \|\theta\|^2 - 2\alpha\theta^T\xi - 2\alpha\theta^T(\xi(\theta) - \xi((\mathrm{id} - \alpha\xi)^{-1}(\theta))) + O(\alpha^2) \\
&< \|\theta\|^2 - 2\alpha + \alpha + O(\alpha^2) \\
&= \|\theta\|^2 - \alpha(1 - O(\alpha)) < \|\theta\|^2
\end{aligned}$$

for all $\sqrt{3} < \|\theta\| < s$ and $\alpha < \delta$ sufficiently small.

**CO, CGD, LA, LOLA, SOS.** Writing $\nu$ for $\gamma$ if $F = F_{CO}$ and $\nu$ for $\alpha$ otherwise, for each algorithm we have

$$F(\theta) = \theta - \alpha\xi + \alpha\nu K$$

for some continuous function $K : \mathbb{R}^d \to \mathbb{R}$. For instance, $K = -H^T\xi$ for CO (see Appendix A). We obtain

$$\begin{aligned}
\|F(\theta)\|^2 &= \|\theta - \alpha\xi + \alpha\nu K\|^2 \\
&= \|\theta\|^2 - 2\alpha\theta^T\xi + 2\alpha\nu\theta^T K - 2\alpha^2\nu\xi^T K + \alpha^2\|\xi\|^2 + \alpha^2\nu^2\|K\| \\
&= \|\theta\|^2 - \alpha\left(2\theta^T\xi - 2\nu\theta^T K + 2\alpha\nu\xi^T K - \alpha\|\xi\|^2 - \alpha\nu^2\|K\|\right).
\end{aligned}$$

Notice that every term in the brackets contains an $\alpha$ or $\nu$ except for the first. We have already shown that $\theta^T\xi > 1$ for all $\|\theta\|^2 > 3$ for GD above, hence for any $s > 0$ we have

$$\begin{aligned}
\|F(\theta)\|^2 &< \|\theta\|^2 - \alpha\left(2 - 2\nu\sup_{\theta \in B_s}\theta^T K + 2\alpha\nu\inf_{\theta \in B_s}\xi^T K - \alpha\sup_{\theta \in B_s}\|\xi\|^2 - \alpha\sup_{\theta \in B_s}\nu^2\|K\|\right) \\
&= \|\theta\|^2 - \alpha(2 - O(\alpha, \nu)) < \|\theta\|^2
\end{aligned}$$

for all $\sqrt{3} < \|\theta\|^2 < s$ and $\alpha, \nu$ sufficiently small.

**SGA.** The situation differs from the above since parameter $\lambda$ follows an alignment criterion, namely $\lambda = \mathrm{sign}\left(\langle\xi, H^T\xi\rangle\langle A^T\xi, H^T\xi\rangle\right)$, which cannot be made small. First note that

$$\theta^T G_{SGA} = \theta^t\xi + \lambda\theta^T(A^T\xi) = x^4 + y^4 - x^2 - y^2 + \lambda(x^2 + y^2 + x^3y - xy^3).$$

If $\lambda = -1$,

$$\theta^T G_{SGA} = x^4 + y^4 - 2x^2 - 2y^2 - x^3y + xy^3$$

and splitting $x^4 + y^4$ in two yields

$$\frac{x^4 + y^4}{2} - 2x^2 - 2y^2 = \frac{1}{4}\left[(x^2 - y^2)^2 + (x^2 + y^2)(x^2 + y^2 - 8)\right] > 1$$

for $\|\theta\|^2 = x^2 + y^2 > 9$, while

$$\frac{x^4 + y^4}{2} - x^3y + xy^3 = \frac{1}{2}\left[(-x^2 + xy + y^2)^2 + x^2y^2\right] > 0$$

for $\|\theta\| > 0$. Summing the two yields $\theta^T G_{SGA} > 1$ for $\|\theta\|^2 > 9$ and $\lambda = -1$. If $\lambda = 1$,

$$\begin{aligned}
\theta^T G_{SGA} &= x^4 + y^4 + x^3 y - x y^3 \\
&= x^4 + y^4 - 2x^2 - 2y^2 + x^3 y - x y^3 + 2(x^2 + y^2) \\
&\geq x^4 + y^4 - 2x^2 - 2y^2 + x^3 y - x y^3 > 1
\end{aligned}$$

for $\|\theta\|^2 > 9$ by swapping $x$ and $y$ in the $\lambda = -1$ case above. We conclude $\theta^T G_{SGA} > 1$ for $\|\theta\|^2 > 9$ regardless of $\lambda$. For any $s > 0$ we obtain

$$\|F(\theta)\|^2 = \|\theta\|^2 - 2\alpha \theta^T G_{SGA} + \alpha^2 \|G_{SGA}\|^2 < \|\theta\|^2 - \alpha\left(2 - \alpha \|G_{SGA}\|^2\right) < \|\theta\|^2$$

for all $3 < \|\theta\| < s$ and $\alpha < 2/\sup_{\theta \in B_s} G_{SGA}$.

**Bounded iterates: conclusion.**   Now assume as usual that $\theta_0$ is initalised in any bounded region $U$. For each algorithm we have found $r$ such that for any $s > 0$ we have $\|F(\theta)\| < \|\theta\|$ for all $r < \|\theta\| < s$ and $\alpha, \gamma$ sufficiently small. Now pick $r' \geq r$ such that $U \subset B_{r'}$. Define the bounded region
$$V = \{\theta - tG(\theta) \mid t \in [0,1], \theta \in B_{r'}\}.$$
and pick $s \geq r'$ such that $V \subset B_s$. By the above we have $\|F(\theta)\| < \|\theta\|$ for all $r < \|\theta\| < s$ and $\alpha, \gamma$ sufficiently small. In particular, fix any $\alpha, \gamma < 1$ satisfying this condition. We claim that $F(\theta) \in B_s$ for all $\theta \in B_s$. Indeed, either $\theta \in B_r$ implies $F(\theta) = \theta - \alpha G(\theta) \in V \subset B_s$ or $\theta \notin B_r$ implies $\|F(\theta)\| < \|\theta\| < s$ and so $F(\theta) \in B_s$. We conclude that $\theta_0 \in U \subset B_s$ implies bounded iterates $\theta_k = F^k(\theta) \in B_s$ for all $k$.

**Non-convergence: strategy.**   We show that all methods in $\mathcal{A}$ have the origin as unique fixed points for $\alpha, \gamma$ sufficiently small. Fixed points of each gradient-based method are given by $G = 0$, where $G$ is given in Appendix A, and we moreover show that the Jacobian $\nabla G$ at the origin is negative-definite. Non-convergence will follow from this for $\alpha$ sufficiently small.

**GD.**   Fixed points of simultaneous GD correspond by definition to critical points:
$$G_{\text{GD}} = \xi = 0 \iff \theta = 0.$$
The Jacobian of $G$ at $0$ is
$$\nabla \xi = H = \begin{pmatrix} -1 & 1 \\ -1 & -1 \end{pmatrix} \prec 0.$$

**AGD.**   We have
$$G_{\text{AGD}} = 0 \iff \begin{cases} \xi_1 = 0 \\ \xi_2(\theta_1 - \alpha\xi_1, \theta_2) = 0 \end{cases} \iff \begin{cases} \xi_1 = 0 \\ \xi_2 = 0 \end{cases} \iff \xi = 0 \iff \theta = 0.$$

Now
$$\begin{aligned}
\xi_2(x - \alpha\xi_1(x,y), y) &= -(x - \alpha(y - x + x^3)) - y + y^3 \\
&= x(-1 - \alpha) + y(-1 + \alpha) + \alpha x^3 + y^3
\end{aligned}$$

so the Jacobian at the origin is
$$J_{AGD} = \begin{pmatrix} -1 & 1 \\ -1 - \alpha & -1 + \alpha \end{pmatrix}$$

with symmetric part
$$S_{AGD} = \begin{pmatrix} -1 & -\alpha/2 \\ -\alpha/2 & -1 + \alpha \end{pmatrix}$$

which has negative trace for all $\alpha < 2$ and positive determinant
$$-\alpha^2/2 - \alpha + 1 = -(\alpha+1)^2/2 + 3/2 > -9/8 + 3/2 > 0$$

for all $\alpha < 1/2$, which together imply negative eigenvalues and hence $S_{AGD} \prec 0$. Recall that a matrix is negative-definite iff its symmetric part is, hence $J_{AGD} \prec 0$ for all $\alpha < 1/2$.

**EG.** We have

$$G_{\text{EG}} = \xi \circ (\text{id} - \alpha \xi) = 0 \iff \text{id} - \alpha \xi = 0 \iff \begin{cases} x - \alpha(y - x + x^3) = 0 \\ y - \alpha(-x - y + y^3) = 0. \end{cases}$$

We have shown that any bounded initialisation results in bounded iterates for EG for $\alpha$ sufficiently small. Let $U$ be this bounded region and assume for contradiction that $\text{id} - \alpha\xi = 0$ with $x, y \neq 0$ (noting that $x = 0$ implies $y = 0$ by the first equation and vice-versa). We can assume $x, y > 0$ since any other solution can be obtained by a quadrant change of variable (†). We first prove that $x, y < 1$ for $0 < \alpha < 1/\sup_{\theta \in U}\{y - x + x^3\}$. Indeed we have

$$0 = \xi_1 > x - \alpha \sup_{\theta \in U} > x - 1$$

hence $x < 1$. A similar derivation holds for $y$, hence $0 < x, y < 1$. But now $x \geq y$ implies

$$0 = \xi_1 \geq x - \alpha(y - y + x^3) = x(1 - \alpha x^2) \geq x(1 - \alpha) > 0$$

for $\alpha < 1$ while $x < y$ implies

$$0 = \xi_2 \geq y - \alpha(-x - x + y^3) = y(1 - \alpha y^2) \geq y(1 - \alpha) > 0$$

and the contradiction is complete, hence $\theta = 0$ is the only fixed point of EG. Now

$$J_{EG} = H(I - \alpha H) = \begin{pmatrix} -1 & 1 \\ -1 & -1 \end{pmatrix} \begin{pmatrix} 1 + \alpha & -\alpha \\ \alpha & 1 + \alpha \end{pmatrix} = \begin{pmatrix} -1 & 1 + 2\alpha \\ -1 - 2\alpha & -1 \end{pmatrix}$$

with $S_{EG} = -I \prec 0$, hence $J_{EG} \prec 0$ for all $\alpha$.

**OMD.** By Daskalakis & Panageas (2018, Remark 1.5), fixed points of OMD must satisfy $\xi = 0$ by viewing OMD as mapping pairs $(\theta_k, \theta_{k-1})$ to pairs $(\theta_{k+1}, \theta_k)$, hence $\theta = 0$. Now

$$J_{OMD} = 2H - H(I - \alpha H)^{-1} = 2 \begin{pmatrix} -1 & 1 \\ -1 & -1 \end{pmatrix} - \frac{1}{1 + 2\alpha + 2\alpha^2} \begin{pmatrix} -1 - 2\alpha & 1 \\ -1 & -1 - 2\alpha \end{pmatrix}.$$

Now notice that

$$\frac{1 + 2\alpha}{1 + 2\alpha + 2\alpha^2} \leq 1$$

and so

$$S_{OMD} = \begin{pmatrix} -2 + \frac{1 + 2\alpha}{1 + 2\alpha + 2\alpha^2} & 0 \\ 0 & -2 + \frac{1 + 2\alpha}{1 + 2\alpha + 2\alpha^2} \end{pmatrix} \prec 0$$

for all $\alpha$.

**CO.** We have

$$G_{\text{CO}} = (I + \gamma H^T)\xi = 0 \iff \xi = 0 \iff \theta = 0$$

for all $\gamma$ since the matrix

$$(I + \gamma H^T) = \begin{pmatrix} 1 - \gamma & -\gamma \\ \gamma & 1 - \gamma \end{pmatrix}$$

is always invertible with determinant $(1 - \gamma)^2 + \gamma^2 > 0$. Now

$$J_{CO} = (I + \gamma H^T)H = \begin{pmatrix} 1 - \gamma & -\gamma \\ \gamma & 1 - \gamma \end{pmatrix} \begin{pmatrix} -1 & 1 \\ -1 & -1 \end{pmatrix} = \begin{pmatrix} -1 + 2\gamma & 1 \\ -1 & -1 + 2\gamma \end{pmatrix} \prec 0$$

for all $\gamma < 1/2$.

**SGA.** We have
$$G_{\text{SGA}} = (I + \lambda A^T)\xi = 0 \iff \xi = 0 \iff \theta = 0$$
since antisymmetric $A$ with eigenvalues $ia$, $a \in \mathbb{R}$ implies that $I + \lambda A^T$ is always invertible with eigenvalues $1 + i\lambda a \neq 0$. Now recall that $\lambda$ is given by
$$\lambda = \text{sign}\left(\langle \xi, H^T \xi \rangle \langle A^T, H^T \xi \rangle\right) = \text{sign}\left(\xi^T H^T \xi \cdot \xi^T A H^T \xi\right).$$

We have
$$H^T = \begin{pmatrix} -1 + 3x^2 & -1 \\ 1 & -1 + 3y^2 \end{pmatrix} \prec 0$$

and
$$AH^T = \begin{pmatrix} 1 & -1 + 3y^2 \\ 1 - 3x^2 & 1 \end{pmatrix} \succ 0$$

for all $\|\theta\|$ sufficiently small, hence $\xi^T H^T \xi \leq 0$ and $\xi^T A H^T \xi \geq 0$ and thus
$$\lambda = \text{sign}\left(\langle \xi, H^T \xi \rangle \langle A^T, H^T \xi \rangle\right) = \text{sign}\left(\xi^T H^T \xi \cdot \xi^T A H^T \xi\right) \leq 0$$

around the origin. Now
$$J_{SGA} = (I + \lambda A^T)H = \begin{pmatrix} 1 & -\lambda \\ \lambda & 1 \end{pmatrix}\begin{pmatrix} -1 & 1 \\ -1 & -1 \end{pmatrix} = \begin{pmatrix} -1 + \lambda & 1 + \lambda \\ -1 - \lambda & -1 + \lambda \end{pmatrix} \prec 0$$

for all $\lambda < 1$, which holds in particular for $\lambda \leq 0$.

**CGD.** Note that
$$H_o = \begin{pmatrix} 0 & 1 \\ -1 & 0 \end{pmatrix} = A$$

is antisymmetric, hence $I + \alpha H_o$ is always invertible as for SGA and
$$G_{\text{CGD}} = (I + \alpha H_o)^{-1}\xi = 0 \iff \xi = 0 \iff \theta = 0.$$

Now
$$J_{CGD} = (I + \alpha H_o)^{-1}H = \frac{1}{1 + \alpha^2}\begin{pmatrix} 1 & -\alpha \\ \alpha & 1 \end{pmatrix}\begin{pmatrix} -1 & 1 \\ -1 & -1 \end{pmatrix} = \frac{1}{1 + \alpha^2}\begin{pmatrix} -1 + \alpha & 1 + \alpha \\ -1 - \alpha & -1 + \alpha \end{pmatrix} \prec 0$$

for all $\alpha < 1$.

**LA.** As above,
$$G_{\text{LA}} = (I - \alpha H_o)\xi = 0 \iff \xi = 0 \iff \theta = 0$$
since $(I - \alpha H_o)$ is always invertible. Now
$$J_{LA} = (I - \alpha H_o)H = (I - \alpha A)H = \begin{pmatrix} -1 + \alpha & 1 + \alpha \\ -1 - \alpha & -1 + \alpha \end{pmatrix} \prec 0$$

for all $\alpha < 1$.

**LOLA.** Notice that
$$\text{diag}\left(H_o^T \nabla L\right) = \text{diag}\left(\begin{pmatrix} 0 & -1 \\ 1 & 0 \end{pmatrix}\begin{pmatrix} y - x + x^3 & -y + x - x^3 \\ x + y - y^3 & -x - y + y^3 \end{pmatrix}\right)$$
$$= \begin{pmatrix} -x - y + y^3 \\ -y + x - x^3 \end{pmatrix} = H_o \xi$$

and so
$$G_{\text{LOLA}} = (I - \alpha H_o)\xi - \alpha \, \text{diag}\left(H_o^T \nabla L\right) = (I - 2\alpha H_o)\xi \iff \xi = 0 \iff \theta = 0$$

as for LA. Similarly, substituting $2\alpha$ for $\alpha$ in the derivation for LA yields
$$J_{LOLA} = (I - 2\alpha H_o)H \prec 0$$

for all $\alpha < 1/2$.

**SOS.** As for LOLA we have

$$G_{\text{SOS}} = (I - \alpha H_o)\xi - p\alpha \operatorname{diag}\left(H_o^T \nabla L\right) = (I - \alpha(1+p)H_o)\xi \iff \xi = 0 \iff \theta = 0$$

for any $\alpha, p$. Now $p(\bar{\theta}) = 0$ for fixed points $\bar{\theta}$ by Letcher et al. (2019b, Lemma D.7), hence

$$J_{SOS} = J_{LA} = \begin{pmatrix} -1 + \alpha & 1 + \alpha \\ -1 - \alpha & -1 + \alpha \end{pmatrix} \prec 0$$

for all $\alpha < 1$.

**Non-convergence: conclusion.** We conclude that all algorithms in $\mathcal{A}$ have the origin as unique fixed points, with negative-definite Jacobian, for $\alpha, \gamma$ sufficiently small. If a method converges, it must therefore converge to the origin. We show that this occurs with zero probability. One may invoke the Stable Manifold Theorem from dynamical systems, but there is a more direct proof.

Take any algorithm $F$ in $\mathcal{A}$ and let $U$ be the initialisation region. We prove that the stable set

$$Z = \{\theta_0 \in U \mid \lim_k F^k(\theta_0) = 0\}$$

has Lebesgue measure zero for $\alpha$ sufficiently small. First assume for contradiction that $\theta_k \to 0$ with $\theta_k \neq 0$ for all $k$. Then

$$G(\theta_k) = G(0) + \nabla G(0)\theta_k + O(\|\theta_k\|^2) = \nabla G(\bar{\theta})(\theta_k) + O(\|\theta_k\|^2)$$

since $G(0) = 0$, and we obtain

$$\begin{aligned}
\|\theta_{k+1}\|^2 &= \|\theta_k - \alpha G(\theta_k)\|^2 \\
&= \|\theta_k\|^2 - 2\alpha \theta_k^T G(\theta_k) + \alpha^2 \|G(\theta_k)\|^2 \\
&\geq \|\theta_k\|^2 - 2\alpha \theta_k^T \nabla G(0)\theta_k + O(\|\theta_k\|^3) > \|\theta_k\|^2
\end{aligned}$$

for all $k$ sufficiently large, since $\nabla G(0) \prec 0$. This is a contradiction to $\theta_k \to 0$, so $\theta_k \to 0$ implies $\theta_k = 0$ for some $k$ and so, writing $F_U : U \to \mathbb{R}^d$ for the restriction of $F$ to $U$,

$$Z \subset \cup_{k=0}^{\infty} F_U^{-k}(\{0\}).$$

We claim that $F_U$ is a $C^1$ local diffeomorphism, and a diffeomorphism onto its image. Now $G_U$ is $C^1$ with bounded domain, hence $L$-Lipschitz for some finite $L$. By Lemma 0, the eigenvalues of $\nabla G$ in $U$ satisfy $|\lambda| \leq \|\nabla G\| \leq L$, hence $\nabla F_U = I - \alpha \nabla G_U$ has eigenvalues $1 - \alpha\lambda \geq 1 - \alpha|\lambda| \geq 1 - \alpha L > 0$. It follows that $\nabla F_U$ is invertible everywhere, so $F_U$ is a local diffeomorphism by the Inverse Function Theorem (Spivak, 1971, Th. 2.11). To prove that $F_U : U \to F(U)$ is a diffeomorphism, it is sufficient to show injectivity of $F_U$. Assume for contradiction that $F_U(\theta) = F_U(\theta')$ with $\theta \neq \theta'$. Then by definition,

$$\theta - \theta' = \alpha(G_U(\theta') - G_U(\theta))$$

and so

$$\|\theta - \theta'\| = \alpha \|G_U(\theta') - G_U(\theta)\| \leq \alpha L \|\theta - \theta'\| < \|\theta - \theta'\|,$$

a contradiction. We conclude that $F_U$ is a diffeomorphism onto its image with continuously differentiable inverse $F_U^{-1}$, hence $F_U^{-1}$ is locally Lipschitz and preserves measure zero sets. It follows by induction that $\mu(F_U^{-k}(\{0\})) = 0$ for all $k$, and so

$$\mu(Z) \leq \mu\left(\cup_{k=0}^{\infty} F_U^{-k}(\{0\})\right) = 0$$

since countable unions of measure zero sets have zero measure. Since $\theta_0$ follows a continuous distribution $\nu$, we conclude

$$P_\nu\left(\lim_k F^k(\theta_0) = 0\right) = 0$$

as required. Since all algorithms were also shown to produce bounded iterates, they almost surely have bounded non-convergent iterates for $\alpha, \gamma$ sufficiently small. The proof is complete; iterates are plotted for a single run of each algorithm in Figure 3 with $\alpha = \gamma = 0.01$. $\qquad\square$

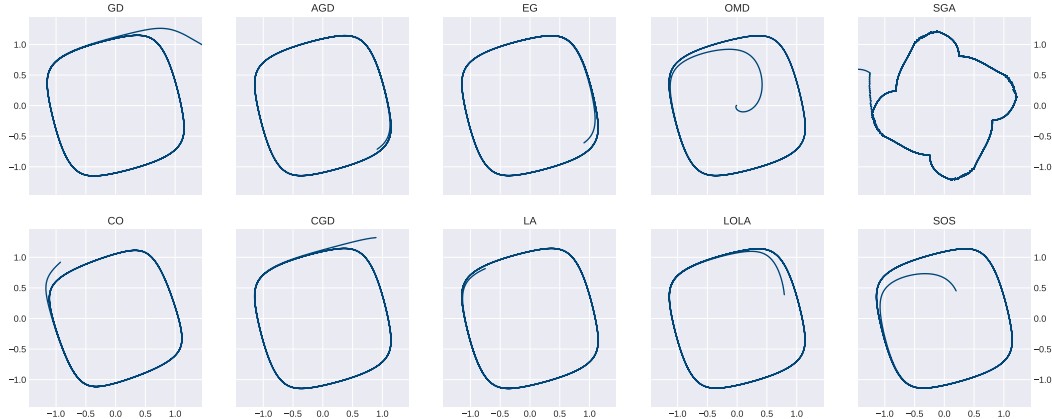

Figure 3: Algorithms in $\mathcal{A}$ fail to converge in $\mathcal{N}$ with $\alpha = \gamma = 0.01$. Single run with standard normal initialisation, 3000 iterations.

## G  PROOF OF COROLLARY 1

**Corollary 1.** *There are no measures of progress for reasonable algorithms which produce bounded iterates in $\mathcal{M}$ or $\mathcal{N}$.*

*Proof.* Assume for contradiction that a measure of progress $M$ exists for some reasonable algorithm $F$ and consider the iterates $\theta_k$ produced in the game $\mathcal{M}$ or $\mathcal{N}$. We prove that the set of accumulation points of $\theta_k$ is a subset of critical points, following Lange (2013, Prop. 12.4.2). Consider any accumulation point $\bar{\theta} = \lim_{m \to \infty} \theta_{k_m}$. The sequence $M(\theta_k)$ is monotonically decreasing and bounded below, hence convergent. In particular,

$$\lim_m M(F(\theta_{k_m})) = \lim_m M(\theta_{k_m+1}) = \lim_m M(\theta_{k_m}).$$

By continuity of $M$ and $F$, we obtain

$$M(F(\bar{\theta})) = M(\lim_m F(\theta_{k_m})) = \lim_m M(F(\theta_{k_m})) = \lim_m M(\theta_{k_m}) = M(\bar{\theta})$$

and hence $F(\bar{\theta}) = \bar{\theta}$. Since $F$ is reasonable, $\bar{\theta}$ must be a critical point. Now the only critical point of $\mathcal{M}$ or $\mathcal{N}$ is the strict maximum $\bar{\theta} = 0$, so any accumulation point of $\theta_k$ must be $\bar{\theta}$. The sequence $\theta_k$ is assumed to be bounded, so it must have at least one accumulation point by Bolzano-Weierstrass. A sequence with exactly one accumulation point is convergent, hence $\theta_k \to \bar{\theta}$. This is in contradiction with the algorithm being reasonable. $\quad\square$

