# OpenReview forum: "On the Impossibility of Global Convergence in Multi-Loss Optimization"
_ICLR.cc/2021/Conference — ICLR 2021 Poster_

### Official Review · AnonReviewer1 · 2020-10-28
**Meaningful Result**

**Rating:** 8
**Confidence:** 5

**Review:**

This paper shows that the class of two-player markets have no satisfactory outcome in the usual sense. Players should neither escape to infinite losses nor converge to strict maxima or non-critical point. Some concrete examples are analyzed with negative results.

This paper is a reminder of researchers: we should carefully model the objective functions of multiple interacting intelligent agents and the interactions between them.

Weakness:
It is better to describe the \alpha and \gamma in Sec. 3.3.

---

> ### Author Response · Authors · 2020-11-19
> **Thank you.**
>
> Thank you for your encouraging review. We have updated the paper to describe $\alpha$ and $\gamma$ in Section 3.3.

---

### Official Review · AnonReviewer3 · 2020-10-29
**A well written paper providing an important negative result regarding multi-objective optimization**

**Rating:** 7
**Confidence:** 4

**Review:**

## Summary:

This paper deals with the impossibility of global convergence to stationary points in multi-loss optimization. The authors introduce some problems for which any “reasonable“ method has an undesired behavior.
The notion of “reasonable method” is quite general and makes the result of this paper interesting.


### pros:
- The framework of “reasonable methods” is quite general.
- The paper answers an important question of the game optimization community.
- The paper is easy to read and well written.

### cons:
- there are some imprecisions (see my section about questions)


## Questions:
The questions are asked by decreasing order of importance.

### Proof of Theorem 2

In the proof of Theorem 2 (page 19) you claim that the function $h(\theta):= \inf_k (\|F_0^k(\theta)\|)$ is continuous. I do not know if it is true or false (it is likely to be true), but this claim is not obvious (since it is an infimum of continuous functions).

Also, the statement in your theorems only considers the reasonable algorithms with the hyperparameters, ensuring that (R2) is valid. An easy way to get rid of this additional assumption may be to remove the mention of hyperparameter (in the sense that an algorithm with hyperparameters that make it locally converge to local maxima is not a reasonable algorithm).

### Definition of $F(\theta)$
At the end of page 3, you have $F$ that can depend on all the previous iterates $F(\theta_k,\ldots,\theta_0)$ but then in your definition R1 $F$ only depend on a single iterate $\theta$.
How can you reconcile that a method may depend on all the previous iterates and Assumption R1? One simple way is maybe to consider a time-dependent operator $F_k$ such that $F_k(\theta_0) = \theta_{k+1}$.  .

### About Proposition 2

It seems to me that most of the results of this proposition have been already covered by
 [Daskalakis and Panageas, 2018] [Adolphs et al., 2018] [Mazumdar et al. 2020] and[Berard et al. 2020]. What is your relative contribution?
Also, note that $H_d$ and $S$ are not defined.

### Global convergence in Balduzzi et al. 2018
I am not sure what global convergence result you are mentioning. Also, in your related work section, you should differentiate Hsieh et al. 2020 from the other related work. While  Hsieh et al. 2020 is trying to tackle the general nonconvex-nonconcave (which is very related to your work), the other related work mentioned in the first paragraph of Page 2 consider restrictive assumptions that allow them to get a Lyapunov function related to convex optimization (while your work implies that such a function does not exist in general).



### Minor comment
- the sentence “Lipschitz continuity of $\nabla f$ is not assumed, which would fail even for cubic polynomials” is a bit confusing since you mention the Lipschitz constant of $\nabla f$ in Proposition 1. I suggest emphasizing that the *global* (outside of $U_0$) Lipschitz continuity is not assumed.
- In the contribution section, $\mathcal A$ is not defined. (I suggest to point to the appendix or specify that $\mathcal A$ is the restricted class of function you mentioned at the beginning of your sentence)
- Note that it would be interesting to discuss the Lyapunov function used to show the global convergence in the non-convex minimization setting (the suboptimality f(\theta) -f^*). The current illustration you have  (page 8) also works in the multi-objective setting (by considering that H has a positive definite symmetric part). Thus, it is not very insightful of the main difference between single and multi-objective minimization.

---

> ### Author Response · Authors · 2020-11-19
> **We have updated the paper to incorporate your questions/comments.**
>
> Thank you for your detailed review. We address each question below.
>
> ### Proof of Theorem 2
> Thanks for pointing this out: $h$ is upper semi-continuous but not necessarily continuous. We have corrected and clarified the proof in the uploaded revision.
>
> We thought about defining algorithms to be reasonable if they avoid strict maxima for ALL hyperparameters upon writing the paper, but chose to avoid this stronger requirement. For instance, Consensus Optimization (CO) is reasonable for $\gamma$ sufficiently small because strict maxima of the game remain unstable with respect to CO dynamics. However, CO may well converge to strict maxima if $\gamma$ is too large; it seems acceptable for algorithms to behave well only in some hyperparameter range. This is similar to gradient descent behaving well only for appropriately small learning rates: it may otherwise diverge to infinity, even for coercive losses.
>
> ### Definition of $F(\theta)$
> We can easily reconcile this abuse of notation by defining (R1) as $F(\theta_k, \ldots, \theta_0) = \theta_k \implies \xi(\theta_k) = 0$. This is consistent with the intuition that a non-zero gradient at current parameters $T_k$ should lead to further decrease of the loss for any reasonable algorithm $F$, regardless of previous parameter dependence.
>
> ### About Proposition 2
> The proposition is rather trivial to prove and can indeed be seen as a collage of existing results, but is included to point clearly (and more visually than can be found elsewhere) to the fact that $H(\theta) \prec 0$ is the weakest generalisation of the concept of ‘strict maxima’ to multi-loss optimization. We do not believe this was established in previous work, though in any case, is not central to our paper. It is simply a further justification that our assumptions on reasonable algorithms are as weak as possible.
>
> ### Global convergence in Balduzzi et al. 2018
> (Balduzzi et al., 2018) establish global convergence of Hamiltonian gradient descent in Hamiltonian games in Section 2.5, Theorem 3, under some regularity conditions. They also state without proof the global convergence of simultaneous gradient descent in potential games in Section 2.4: “[potential games] are games where simultaneous gradient descent on the losses is gradient descent on a single function. It follows that descent on $\xi$ converges to a fixed point that is a local minimum of $\phi$”. Despite the lack of proof, one can indeed establish that simultaneous gradient descent converges globally in potential games by appealing to global convergence of gradient descent on single losses under the usual regularity conditions, as in Proposition 1 of our paper.
>
> We differentiated the work of (Hsieh et al., 2020) from other related work by placing it in a separate in-depth paragraph and stating that their focus is similar to ours, but will clarify the difference further.
>
> ### Minor comment
> We have incorporated these helpful comments in the revision.

---

### Official Review · AnonReviewer2 · 2020-10-30
**Games are not optimization**

**Rating:** 6
**Confidence:** 4

**Review:**

Thank you for the explanation and the update of the writeup.
I still find the main message interesting but not critical for any real application.
Hence my evaluation remains as weak accept.
Thank you.



%%%%%%%%%%%%%%%%%%%%%%%%%%%%%%%%%%%%%%%%%%%%%%%%%%%%%%%%%%


The paper provides an impossibility type of result for convergence of reasonable dynamics in general multi-loss settings. Specifically, the paper constructs a game between two agents whose only simultaneous critical point is a strict maximum. Thus under the definition of a 'reasonable' algorithm as an algorithm that avoids strict maxima, the dynamics cannot converge to equilibria in this case and has to therefore cycle.

The paper builds on a series of recent papers that show that many reasonable dynamics fail to converge in reasonable multi-loss settings (e.g. GD in zero-sum games). On the positive side, the results accounts for a large class of optimization/gradient driven dynamics. Also, the example of the multi-agent setting is relatively small. Two agents/two degrees of freedom. On the negative side, the example is rather artificial and it does not seem to capture any specific setting of independent interest. Moreover, there is little intuition in the current writeup behind the construction of the example. The appendix just shows that this example satisfies the target properties.

The main of message of the paper is that multi-loss settings are very different from single loss settings and we cannot quickly and easily apply tools from optimization and hope to succeed at least not in all cases. The predominance of multi-loss architectures in AI settings makes this a reasonable message for ICLR.  On the other hand, it is clear that we are not moving away from these architectures any time soon, so such a message is less effective than explanation/tools of how to understand multi-loss environments or make them work better.

---

> ### Author Response · Authors · 2020-11-19
> **We have updated the paper with further explanation/intuition.**
>
> Thank you for your review. As you point out, the main message of our paper is that we cannot always hope to succeed in the realm of multi-loss optimization, at least in the traditional sense of converging to some kind of ‘local minimum’. Applied examples of this undesirable phenomenon may appear further down the road. However, a first step to finding them or disproving their existence is to recognize the possibility of cyclic behaviour *in principle*, as established here.
>
> Our example in Theorem 1 was constructed by noticing that there is no necessary reason for the local minima of two coercive losses to coincide, even when these losses are tied by zero-sum interaction terms in only two parameters. Instead, the gradients of each loss may only *simultaneously* vanish at a local maximum in each player’s respective coordinate. Looking more specifically at our construction, the highest-order terms (first and last) provide coercivity in both coordinates while still having zero-sum interactions. The $-x^2, -y^2$ terms yield a strict local maximum *at* the origin, while the $\pm xy$ terms provide opposite incentives *around* the origin which prevent any other simultaneous critical point to arise. We have added this explanation in the appendix of our revised paper.

---

### Official Review · AnonReviewer4 · 2020-11-02
**The paper studies the convergence problem in a two-player game with zero-sum interactions, whose losses are both coercive and analytic. The paper provides some new insights into the studied problem, but the reviewer questions its practical value for the community. Though as the major contribution of the paper, the three developed theorems, as the reviewer thinks, have their respective shortcomings for practical use.**

**Rating:** 4
**Confidence:** 4

**Review:**



1. For Theorem 1, as the reviewer understands it, for an optimization problem whose only critical point is a strict maxima, it only has four outcomes, which are listed in the theorem. The result seems quite intuitive and provides very limited understanding for the problem. Please list other possible outcomes for the general problem and state in such way that the paper finds some impossible outcomes which can be excluded for consideration.
2. Theorem 2 states that a two-player game with a specific loss cannot converge to the only strict minimum. As the major result of the paper, this finding however has very limited value in practice. The specifically designed loss is not justified to be used in reality, and the result is built upon this loss. If the loss has nowhere to be found in use, investigating such a loss inspires the community very little.
3. Similar to Theorem 2, it states in Theorem 3 that a two-player game with a specific loss cannot converge to the only strict maximum. Even more unclear to the reviewer, such a result can provide useful value to the community or not. A game with only strict maximum cannot converge to the maximum is not quite interesting to the community (as the reviewer believes).
Some minor comments:
The paper use the term of “simultaneous” quite a lot in the first part of the paper. This is quite confusing for the readers as this term is not quite acknowledged.

---

> ### Author Response · Authors · 2020-11-19
> **The practical value of our paper is to warn the machine learning community that reasonable optimization algorithms have no global convergence guarantees in multi-loss optimization, contrary to single-loss optimization, even in simple classes of games. Proving or disproving this for specific applications (eg GANs) is an important avenue for further research, but requires that we recognise the impossibility of global guarantees in the first place.**
>
> Thank you for your review. We address each point below.
>
> 1. The obvious desired outcome for a standard minimization problem is to reach a local (if not global) minimum of the loss. Theorem 1 points out that some simple games (two-player, two-parameter markets) do not have such an outcome. None of the four outcomes include converging to a local minimum, because there are none -- despite coercivity, analyticity and nondegeneracy of the losses. This is not intuitive at first sight, or at the very least, was left unnoticed by previous researchers in the field.
>
> 2. As with many theoretical results, applied examples of the phenomenon we exhibit may appear further down the road. However, a first step to finding them or disproving their existence is to recognize the possibility of cyclic behaviour *in principle*, as we establish here. To emphasise the last paragraph of our paper: “The hope for machine learning practitioners is that local minima with large regions of attraction prevent limit cycles from arising in applications of interest, including GANs. Proving or disproving this is an interesting and important avenue for further research.”
>
> 3. It is unclear to us why the reviewer “believes” but does not justify that “a game with only strict maximum cannot converge to the maximum is not quite interesting to the community”, beyond the fact that it makes no sense to say that “a game cannot converge to the maximum”. Theorem 3 instead states that “algorithms in $\mathcal{A}$ almost surely have bounded nonconvergent iterates” in a given weakly-coercive zero-sum game, for appropriate hyperparameters. This was previously unknown to the community, and acts as a warning that even simple zero-sum games may lead to cyclic iterates for a wide class of algorithms.

---

> > ### Comment · AnonReviewer4 · 2020-11-25
> > **All the majors results are built upon specifically designed losses and looks like no one are using these losses in reality**
> >
> > Thanks for the authors' effort of trying to address my comments. However, my concerns still persist and have not been fully addressed. One of the biggest concerns is the practical use of the results in the paper. All the majors results are built upon specifically designed losses and looks like no one are using these losses in reality. I was looking for either of two aspects to address my comments:
> > 1. Current---A justification that the specifically designed losses, upon which the authors have derived their results, are (widely) used in practice. Or perhaps they share common properties with the losses that are used in reality.
> > 2. Future---An analysis of inspiration that even the results are not based on general losses, the results derived can potentially motivate the community for further study.
> >
> > Nevertheless, I was not able to find either of the above two in the paper or from the rebuttal. Essentially, the results have not been justified valuable for current or future use. Therefore, I would like to keep my rating score for this paper at this point.

---

### Decision · Program_Chairs · 2021-01-07
**Final Decision**

**Decision:**

Accept (Poster)

**Comment:**

This paper presents a series of negative results regarding the convergence of deterministic, "reasonable" algorithms in min-max games. The defining characteristic of such algorithms is that (a) the algorithm's fixed points are critical points of the game; and (b) they avoid strict maxima from almost any initialization. The authors then construct a range of simple $2$-dimensional "market games" in which every reasonable algorithm fails to converge, from almost any initialization.

The paper received three positive recommendations and one negative, with all reviewers indicating high confidence. After my own reading of the paper, I concur with the majority view that the paper's message is an interesting one for the community and will likely attract interest in ICLR.

In more detail, I view the authors' result as a cautionary tale, not unlike the NeurIPS 2019 spotlight paper of Vlatakis-Gkaragkounis et al, and a concurrent arxiv preprint by Hsieh et al. (2020). In contrast to the type of cycling/recurrence phenomena that are well-documented in bilinear games (and which can be resolved through the use of extra-gradient methods), the non-convergence phenomena described by the authors of this paper appear to be considerably more resilient, as they apply to all "reasonable" algorithms. Determining whether GANs (or other practical applications of min-max optimization) can exhibit such phenomena is an important open question, and one which needs to be informed by a deeper understanding of the theory. I find this paper successful in this regard and I am happy to recommend acceptance.